# Zero-Order One-Point Gradient Estimate in Consensus-Based Distributed Stochastic Optimization

**Elissa Mhanna**                                                *elissa.mhanna@centralesupelec.fr*
**Mohamad Assaad**                                            *mohamad.assaad@centralesupelec.fr*
*Laboratoire des Signaux & Systèmes*
*CentraleSupélec*
*3 rue Joliot Curie*
*91190 Gif-sur-Yvette, France*

**Reviewed on OpenReview:** *https://openreview.net/forum?id=tkswb7XoYB*

## Abstract

In this work, we consider a distributed multi-agent stochastic optimization problem, where each agent holds a local objective function that is smooth and strongly convex and that is subject to a stochastic process. The goal is for all agents to collaborate to find a common solution that optimizes the sum of these local functions. With the practical assumption that agents can only obtain noisy numerical function queries at precisely one point at a time, we consider an extention of a standard consensus-based distributed stochastic gradient (DSG) method to the bandit setting where we do not have access to the gradient, and we introduce a zero-order (ZO) one-point estimate (1P-DSG). We analyze the convergence of this techniques using stochastic approximation tools, and we prove that it *converges almost surely to the optimum* despite the biasedness of our gradient estimate. We then study the convergence rate of our method. With constant step sizes, our method competes with its first-order (FO) counterparts by achieving a linear rate $O(\varrho^k)$ as a function of number of iterations $k$. To the best of our knowledge, this is the first work that proves this rate in the noisy estimation setting or with one-point estimators. With vanishing step sizes, we establish a rate of $O(\frac{1}{\sqrt{k}})$ after a sufficient number of iterations $k > K_0$. This rate matches the lower bound of centralized techniques utilizing one-point estimators. We then provide a regret bound of $O(\sqrt{k})$ with vanishing step sizes. We further illustrate the usefulness of the proposed technique using numerical experiments.

## 1 Introduction

Gradient-free optimization is an old topic in the research community; however, there has been an increased interest recently, especially in machine learning applications, where optimization problems are typically solved with gradient descent algorithms. Successful applications of gradient-free methods in machine learning include competing with an adversary in bandit problems (Flaxman et al., 2004; Agarwal et al., 2010), generating adversarial attacks for deep learning models (Chen et al., 2019; Liu et al., 2019) and reinforcement learning (Vemula et al., 2019). Gradient-free optimization aims to solve optimization problems with only functional ZO information rather than FO gradient information. These techniques are essential in settings where explicit gradient computation may be impractical, expensive, or impossible. Instances of such settings include high data dimensionality, time or resource straining function differentiation, or the cost function not having a closed-form. ZO information-based methods include direct search methods (Golovin et al., 2019), 1-point methods (Flaxman et al., 2004; Bach & Perchet, 2016; Vemula et al., 2019; Li & Assaad, 2021) where a function $f(\cdot, S) : \mathbb{R}^d \to \mathbb{R}$ is evaluated at a single point with some randomization to estimate the gradient as such

$$g_{\gamma,z}^{(1)}(x, S) = \frac{d}{\gamma} f(x + \gamma z, S) z, \tag{1}$$

with $x$ the optimization variable, $\gamma > 0$ a small value, and $z$ a random vector following a symmetrical distribution. ZO also includes 2- or more point methods (Duchi et al., 2015; Nesterov & Spokoiny, 2017; Gorbunov et al., 2018; Bach & Perchet, 2016; Hajinezhad et al., 2019; Kumar Sahu et al., 2018; Agarwal et al., 2010; Chen et al., 2019; Liu et al., 2019; Vemula et al., 2019), where functional difference at various points is employed for estimation, generally having the respective structures

$$g_{\gamma,z}^{(2)}(x, S) = d\frac{f(x + \gamma z, S) - f(x - \gamma z, S)}{2\gamma}z \tag{2}$$

$$\text{and } g_{\gamma}^{(2d)}(x, S) = \sum_{j=1}^{d} \frac{f(x + \gamma e_j, S) - f(x - \gamma e_j, S)}{2\gamma}e_j \tag{3}$$

where $\{e_j\}_{j=1,\ldots,d}$ is the canonical basis, and other methods such as sign information of gradient estimates (Liu et al., 2019).

Another area of great interest is distributed multi-agent optimization, where agents try to cooperatively solve a problem with information exchange only limited to immediate neighbors in the network. Distributed computing and data storing are particularly essential in fields such as vehicular communications and coordination, data processing and distributed control in sensor networks (Shi & Eryilmaz, 2020), big-data analytics (Daneshmand et al., 2015), and federated learning (McMahan et al., 2017). More specifically, one direction of research integrates (sub)gradient-based methods with a consensus/averaging strategy; the local agent incorporates one or multiple consensus steps alongside evaluating the local gradient during optimization. Hence, these algorithms can tackle a fundamental challenge: overcoming differences between agents' local data distributions.

## 1.1 Problem Description

Consider a set of agents $\mathcal{N} = \{1, 2, \ldots, n\}$ connected by a communication network. Each agent $i$ is associated with a local objective function $f_i(\cdot, S) : \mathcal{K} \to \mathbb{R}$, where $\mathcal{K} \subset \mathbb{R}^d$ is a convex feasible set. The global goal of the agents is to collaboratively locate the decision variable $x \in \mathcal{K}$ that solves the stochastic optimization problem:

$$\min_{x \in \mathcal{K}} \mathcal{F}(x) = \frac{1}{n}\sum_{i=1}^{n} F_i(x) \tag{4}$$

where

$$F_i(x) = \mathbb{E}_S f_i(x, S),$$

with $S \in \mathcal{S}$ denoting an i.i.d. ergodic stochastic process describing uncertainties in the communication system.

We assume that at each time step, agent $i$ can only query the function values of $f_i$ at exactly one point, and can only communicate with its neighbors. Further, we assume that the function queries are noisy $\tilde{f}_i = f_i + \zeta_i$ with $\zeta_i$ some additive noise. Agent $i$ must then employ this query to estimate the gradient of the form $g_i(x, S_i)$.

## 1.2 Function Classes and Gradient Estimate Assumptions

Consider the following five classes of functions:

- The convex class $\mathcal{C}_{cvx}$ containing all functions $f : \mathbb{R}^d \to \mathbb{R}$ that are convex.

- The strongly convex class $\mathcal{C}_{sc}$ containing all functions $f : \mathbb{R}^d \to \mathbb{R}$ that are continuously differentiable and admit a constant $\lambda_f$ such that

$$\langle \nabla f(x) - \nabla f(y), x - y \rangle \geq \lambda_f \|x - y\|^2, \quad \forall x, y \in \mathbb{R}^d.$$

- The Lipschitz continuous class $\mathcal{C}_{lip}$ containing all functions $f : \mathbb{R}^d \to \mathbb{R}$ that admit a constant $L_f$ such that

$$|f(x) - f(y)| \leq L_f \|x - y\|, \quad \forall x, y \in \mathbb{R}^d.$$

- The smooth class $\mathcal{C}_{smo}$ containing all functions $f : \mathbb{R}^d \to \mathbb{R}$ that are continuously differentiable and admit a constant $G_f$ such that

$$\|\nabla f(x) - \nabla f(y)\| \leq G_f \|x - y\|, \quad \forall x, y \in \mathbb{R}^d.$$

- The gradient dominated class $\mathcal{C}_{gd}$ containing all functions $f : \mathbb{R}^d \to \mathbb{R}$ that are differentiable, have a global minimizer $x^*$, and admit a constant $\nu_f$ such that

$$2\nu_f(f(x) - f(x^*)) \leq \|\nabla f(x)\|^2, \quad \forall x \in \mathbb{R}^d.$$

This gradient domination property can be viewed as a nonconvex analogy of strong convexity.

In addition, consider the following assumptions on the gradient estimate:

- A gradient estimate $g$ is said to be unbiased w.r.t. the true gradient $\nabla f$ if for all $x \in \mathbb{R}^d$ and independent $S \in \mathcal{S}$, it satisfies the following equality

$$\mathbb{E}_S[g(x, S)|x] = \nabla f(x).$$

- Otherwise, it is said to be biased and satisfies

$$\mathbb{E}_S[g(x, S)|x] = \nabla f(x) + b(x),$$

with $b(x)$ some bias term.

- A gradient estimate $g$ is said to have bounded variance when for all $x \in \mathbb{R}^d$ and independent $S \in \mathcal{S}$,

$$\mathbb{E}_S[\|g(x, S) - \nabla f(x)\|^2|x] \leq \sigma \text{ for some } \sigma > 0.$$

- Otherwise, when this bound is unknown or does not exist, it is said to have unbounded variance.

In general, FO stochastic gradient estimates are unbiased and have bounded variance. ZO estimates, on the other hand, are biased. While multi-point ZO estimates have bounded or even vanishing variance, one-point estimates have unbounded variance Liu et al. (2020).

### 1.3 Related Work

**FO Consensus-Based Distributed Methods:** The optimal convergence rate for solving problem (4), assuming the objective function $\mathcal{F}$ is strongly convex with Lipschitz continuous gradients, has been established as $O(\frac{1}{k})$ under a diminishing step size with full gradient information Pu & Nedić (2018); Nemirovski et al. (2009). However, when employing a constant step size $\alpha > 0$ that is sufficiently small, the iterates produced by a stochastic gradient method converge exponentially fast (in expectation) to an $O(\alpha)$-neighborhood of the optimal solution (Pu & Nedić, 2018); this is known as the linear rate $O(\varrho^k)$. The literature dedicated to solving problem (4) is vast. In what follows, we highlight some of the contributions.

Towfic et al. (2016); Tu & Sayed (2012) study distributed stochastic gradient methods where they compare the adapt-then-combine (ATC) and combine-then-adapt (CTA) strategies, and prove that the ATC strategy outperforms CTA one in terms of convergence rate, whether with vanishing or with constant step sizes and that it is more robust against data distribution drifts and network topology. Jakovetic et al. (2018) consider the CTA strategy with noisy FO gradients over random networks and establish an $O(\frac{1}{k})$ convergence rate for strongly convex and smooth objectives and vanishing step size. Matei & Baras (2011); Yuan et al. (2016) also consider random networks and solve problem (4) using a noise-free (sub)gradient instead and achieve a

| | ESTIMATE | OP | FUNCTION CLASS | STEP SIZE | REGRET BOUND | CONVERGENCE RATE |
|---|---|---|---|---|---|---|
| ZO | One-point | Centralized | $\mathcal{C}_{cvx} \bigcap \mathcal{C}_{lip}$ | f. | $O(k^{\frac{3}{4}})$ | $O(\frac{1}{\sqrt[4]{k}})$ Flaxman et al. (2004) |
| | | Centralized | $\mathcal{C}_{sc} \bigcap \mathcal{C}_{lip} \bigcap \mathcal{C}_{smo}$ | v. | $O(\sqrt{k})$ | $O(\frac{1}{\sqrt{k}})$ Bach & Perchet (2016) |
| | | Distributed | $\mathcal{C}_{sc} \bigcap \mathcal{C}_{smo}$ | v. | $O(\sqrt{k})$ | $O(\frac{1}{\sqrt{k}})$ 1P-DSG |
| | | Distributed | $\mathcal{C}_{sc} \bigcap \mathcal{C}_{smo}$ | f. | - | $O(\varrho^k)$ 1P-DSG |
| | Two-point | Centralized | $\mathcal{C}_{cvx} \bigcap \mathcal{C}_{lip}$ | v. | $O(\sqrt{k})$ | $O(\frac{1}{\sqrt{k}})$ Agarwal et al. (2010) |
| | | Centralized | $\mathcal{C}_{sc} \bigcap \mathcal{C}_{lip}$ | v. | $O(\log k)$ | $O(\frac{\log k}{k})$ Agarwal et al. (2010) |
| | | Distributed | $\mathcal{C}_{lip} \bigcap \mathcal{C}_{smo}$ | v. | - | $O(\frac{1}{\sqrt{k}} \log k)$ Tang et al. (2021) |
| | | Distributed | $\mathcal{C}_{smo} \bigcap \mathcal{C}_{gd}$ | v. | - | $O(\frac{1}{k})$ Tang et al. (2021) |
| | 2d-point | Distributed | $\mathcal{C}_{smo}$ | f. | - | $O(\frac{1}{k})$ Tang et al. (2021) |
| | | Distributed | $\mathcal{C}_{smo} \bigcap \mathcal{C}_{gd}$ | f. | - | $O(\varrho^k)$ Tang et al. (2021) |
| | | Distributed | $\mathcal{C}_{sc} \bigcap \mathcal{C}_{smo}$ | v. | - | $O(\frac{1}{\sqrt{k}})$ Kumar Sahu et al. (2018) |
| FO | Unbiased /BV | Distributed | $\mathcal{C}_{sc} \bigcap \mathcal{C}_{smo}$ | f. | - | $O(\varrho^k)$ Matei & Baras (2011); Yuan et al. (2016); Pu & Nedić (2018) |
| | | Distributed | $\mathcal{C}_{sc} \bigcap \mathcal{C}_{smo}$ | v. | - | $O(\frac{1}{k})$ Jakovetic et al. (2018); Pu & Nedić (2018) |

Table 1: Convergence rates for various algorithms related to our work, classified according to the nature of the gradient estimate, whether the optimization problem (OP) is centralized or distributed, the assumptions on the objective function, whether the step size is fixed (f.) or varying (v.), and the achieved regret bound and convergence rate

linear rate to a neighborhood of the optimum with constant step sizes. Nedić & Olshevsky (2016) consider time-varying and directed networks and present a subgradient-push method based on noisy FO gradients that achieves an $O(\frac{\ln k}{k})$ rate under the same assumptions on the objective function and vanishing step size. Both the works of Shi et al. (2015) and Qu & Li (2018) consider a static version of the objective function and propose methods that employ history information of the gradient. They both obtain a rate of $O(\frac{1}{k})$ for general convex and smooth objectives with constant step sizes. Under the further strong convexity assumption, the static nature of the objective allows them to establish a linear convergence rate to the exact solution instead of a neighborhood of it. Qu & Li (2018) inspire the vast literature on gradient tracking extended to the stochastic setting (Pu & Nedić, 2018; Pu, 2020; Xin et al., 2019) that utilizes local auxiliary variables to track the average of all agents' gradients, the linear rate, however, is established to a neighborhood of the optimum.

**ZO Centralized Methods:** ZO methods are known to have worse convergence rates than their FO counterparts under the same conditions. For example, under a convex centralized setting, Flaxman et al. (2004) prove a regret bound of $O(k^{\frac{3}{4}})$ (or equivalently a rate of $O(\frac{1}{\sqrt[4]{k}})$) with a one-point estimator for Lipschitz continuous functions. For strongly convex and smooth objective functions, Hazan & Levy (2014) and Ito (2020) improve upon this result by proving a regret of $O(\sqrt{k \log k})$ and Bach & Perchet (2016) that of $O(\sqrt{k})$. In the work of Agarwal et al. (2010), when the number of points is two, they prove regret bounds of $\tilde{O}(\sqrt{k})$ with high probability and of $O(\log(k))$ in expectation for strongly convex loss functions. When the number is $d+1$ point, they prove regret bounds of $O(\sqrt{k})$ and of $O(\log(k))$ with strong convexity. The reason why the performance improves with the addition of number of points in the estimate, is that their variance can be bounded, unlike one-point estimates whose variance cannot be bounded (Liu et al., 2020). However, when the function queries are subjected to noise, multi-point estimates start behaving like one-point ones. In noisy function queries (centralized) scenario, it has been proven that gradient-free methods cannot achieve a better

convergence rate than $\Omega(\frac{1}{\sqrt{k}})$ which is the lower bound derived by Duchi et al. (2015); Jamieson et al. (2012); Shamir (2013) for strongly convex and smooth objective functions. In the work of Bubeck et al. (2021), a kernelized loss estimator is proposed where a generalization of Bernoulli convolutions is adopted, and an annealing schedule for exponential weights is used to control the estimator's variance in a focus region for dimensions higher than 1. Their method achieves a regret bound of $O(\sqrt{k})$.

**ZO Consensus-Based Distributed Methods:** In distributed settings, Tang et al. (2021) develop two algorithms for a noise-free nonconvex multi-agent optimization problem aiming at consensus. One of them is gradient-tracking based on a 2d-point estimator of the gradient with vanishing variance that achieves a rate of $O(\frac{1}{k})$ with smoothness assumptions and a linear rate for an extra $\nu$-gradient dominated objective assumption and for fixed step sizes. The other is based on a 2-point estimator following an ATC strategy instead of gradient tracking and achieves a rate of $O(\frac{1}{\sqrt{k}} \log k)$ under Lipschitz continuity and smoothness conditions and $O(\frac{1}{k})$ under an extra gradient dominated function structure. In the nonconvex setting, a gradient-tracking method is also proposed, but with a one-point estimator (Mhanna & Assaad, 2023) where a convergence rate of $O(\frac{1}{\sqrt[3]{k}})$ is established with Lipschitz continuous and smooth functions. Kumar Sahu et al. (2018) propose a standard CTA method where they consider a 2d-point estimate with noisy function queries over random networks. Under smoothness and strong convexity, they establish an $O(\frac{1}{\sqrt{k}})$ convergence rate with vanishing step sizes. Wan et al. (2020; 2022) propose a projection-free method with one-point gradient estimate where a linear optimization step is performed instead of projection. They prove a regret bound of $O(k^{\frac{3}{4}})$ for convex losses and an improved regret of $O(k^{\frac{2}{3}}(\log k)^{\frac{1}{3}})$ for strongly convex ones.

We highlight some of the mentioned convergence rates from the literature in Table 1.

## 1.4 Contributions

While consensus-based distributed methods have been extended to the ZO case (Tang et al., 2021; Kumar Sahu et al., 2018), their approach relies on a multi-point gradient estimator and in the case of Mhanna & Assaad (2023); Wan et al. (2020; 2022), the rates established for one-point estimates are slow. The multi-point estimation technique assumes the ability to observe multiple instances of the objective function under identical system conditions, i.e., many function queries are done for the same realization of $S$ in (2) and (3). However, this assumption needs to be revised in applications such as mobile edge computing (Mao et al., 2017; Chen et al., 2021; Zhou et al., 2022) where computational tasks from mobile users are offloaded to servers within the cellular network. Thus, queries requested from the servers by the users are subject to the wireless environment and are corrupted by noise not necessarily additive. Other applications include sensor selection for an accurate parameter estimation (Liu et al., 2018) where the observation of each sensor is continuously changing. Thus, in such scenarios, one-point estimates offer a vital alternative to solving online optimization/learning problems. Yet, one-point estimators are not generally used because of their slow convergence rate. The main reason is due to their unbounded variance. To avoid this unbounded variance, in this work, we don't use the estimate given in (1), we extend the one point approach in Li & Assaad (2021)'s work where the action of the agent is a scalar and different agents have different variables, to our consensual problem with vector variables. The difference is that in our gradient estimate, we don't divide by $\gamma$. This brings additional challenges in proving that our algorithm converges and a consensus can be achieved by all agents. And even with bounded variance, there's still a difficulty achieving good (linear) convergence rates with two-point estimates due to the constant upper bound of the variance (Tang et al., 2021). Here, despite this constant bound, we were able to go beyond two-point estimates to achieve a linear rate. Moreover, while it requires $2d$ points *with* the gradient tracking method to achieve a linear rate in Tang et al. (2021)'s work, which is twice the dimension of the gradient itself, here we only need one scalar point or query. This is much more computationally efficient. We further replace the gradient tracking method by a standard ATC strategy which is more communication efficient as it requires the sharing of only one vector instead of two.

We summarize our contribution in the following points,

- We consider smooth and strongly convex local objectives, and we consider the distributed stochastic gradient method in the case where we do not have access to the gradient in the noisy setting. Under

the realistic assumption that the agent only has access to a single noisy function value at each time without necessarily knowing the form of this function, we propose a one-point estimator in a stochastic framework.

- Naturally, one-point estimators are biased with respect to the true gradient and suffer from high variance (Liu et al., 2020); Despite this, in this work, we analyze and indeed prove the algorithm's convergence *almost surely* with a biased estimate. This convergence is stronger than expected convergence analysis usually established for ZO optimization. We also consider that a stochastic process influences the objective function from one iteration to the other, which provides a practical modeling for real-world scenarios that involve various sources of stochasticity, not necessarily additive noise.

- We then study the convergence rate and we demonstrate that with fixed step sizes, the algorithm achieves a linear convergence rate $O(\varrho^k)$ to a neighborhood of the optimal solution, marking the first instance where this rate is attained in ZO optimization with one-point/two-point estimates and in a noisy query setting, to the best of our knowledge. This linear rate competes with FO methods and even centralized algorithms in terms of convergence speed (Pu & Nedić, 2018).

- When the step-sizes are vanishing, we prove that a rate of $O(\frac{1}{\sqrt{k}})$ is attainable to converge to an exact solution after a sufficient number of iterations $k > K_0$. This rate satisfies the lower bounds achieved by its centralized counterparts in the same derivative-free setting (Duchi et al., 2015; Jamieson et al., 2012; Shamir, 2013).

- We then show that a regret bound of $O(\sqrt{k})$ is achieved for this algorithm.

- Finally, we support our theoretical claims by providing numerical evidence and comparing the algorithm's performance to its FO and centralized counterparts.

The rest of this paper is divided as follow. In subsection 1.5, we present the mathematical notation followed in this paper. In subsection 1.6, we present the main assumptions of our optimization problem. We then describe our gradient estimate followed by the proposed algorithm in subsection 2.1. We then prove the almost sure convergence of our algorithm in subsection 3.1 and study its rate in subsection 3.2 with varying step sizes. In subsection 3.3, we find its regret bound. And in subsection 3.4, we consider the case of fixed step sizes, study the convergence of our algorithm and its rate. Finally, in section 4, we provide numerical evidence and conclude the paper in section 5.

## 1.5 Notation

In all that follows, vectors are column-shaped unless defined otherwise and $\mathbf{1}$ denotes the vector of all entries equal to 1. For two vectors $a$, $b$ of the same dimension, $\langle a, b \rangle$ is the inner product. For two matrices $A$, $B \in \mathbb{R}^{n \times d}$, we define

$$\langle A, B \rangle = \sum_{i=1}^{n} \langle A_i, B_i \rangle$$

where $A_i$ (respectively, $B_i$) represents the $i$-th row of $A$ (respectively, $B$). This matrix product is the Hilbert-Schmidt inner product which is written as $\langle A, B \rangle = \text{tr}(AB^{\mathrm{T}})$. $\|.\|$ denotes the 2-norm for vectors and the Frobenius norm for matrices.

We next let $\Pi_{\mathcal{K}}(\cdot)$ denote the Euclidean projection of a vector on the set $\mathcal{K}$. We know that this projection on a closed convex set $\mathcal{K}$ is nonexpansive (Kinderlehrer & Stampacchia (2000) - Corollary 2.4), i.e.,

$$\|\Pi_{\mathcal{K}}(x) - \Pi_{\mathcal{K}}(x')\| \leq \|x - x'\|, \quad \forall x, x' \in \mathbb{R}^d. \tag{5}$$

We assume that each agent $i$ maintains a local copy $x_i \in \mathcal{K}$ of the decision variable and each agent's local function is subject to the stochastic variable $S_i \in \mathbb{R}^m$. At iteration $k$, the respective values are denoted as $x_{i,k}$ and $S_{i,k}$. Bold notations denote the concatenated version of the variables, i.e.,

$$\mathbf{x} := [x_1, x_2, \ldots, x_n]^T \text{ is of dimension } n \times d \text{ and } \mathbf{S} := [S_1, S_2, \ldots, S_n]^T \text{ of dimension } n \times m.$$

We then define the mean of the decision variable as $\bar{x} := \frac{1}{n}\mathbf{1}^T\mathbf{x}$ whose dimension is $1 \times d$.

We define the gradient of $F_i$ at the local variable $\nabla F_i(x_i) \in \mathbb{R}^d$ and its Hessian matrix $\nabla^2 F_i(x_i) \in \mathbb{R}^{d \times d}$ and we let

$$\nabla F(\mathbf{x}) := [\nabla F_1(x_1), \nabla F_2(x_2), \ldots, \nabla F_n(x_n)]^T \in \mathbb{R}^{n \times d}$$

and

$$\mathbf{g} := g(\mathbf{x}, \mathbf{S}) := [g_1(x_1, S_1), g_2(x_2, S_2), \ldots, g_n(x_n, S_n)]^T \in \mathbb{R}^{n \times d}.$$

We define its mean $\bar{g} := \frac{1}{n}\mathbf{1}^T\mathbf{g} \in \mathbb{R}^{1 \times d}$ and we denote each agent's gradient estimate at time $k$ by $g_{i,k} = g_i(x_{i,k}, S_{i,k})$.

### 1.6 Basic Assumptions

In this subsection, we introduce the fundamental assumptions that ensure the performance of the 1P-DSG algorithm.

**Assumption 1.1.** *(on the graph) The topology of the network is represented by the graph $\mathcal{G} = (\mathcal{N}, \mathcal{E})$ where the edges in $\mathcal{E} \subseteq \mathcal{N} \times \mathcal{N}$ represent communication links. A graph $\mathcal{G}$ is undirected, i.e., $(i, j) \in \mathcal{E}$ iff $(j, i) \in \mathcal{E}$, and connected (there exists a path of links between any two agents).*

$W = [w_{ij}] \in \mathbb{R}^{n \times n}$ *denotes the agents' coupling matrix, where agents $i$ and $j$ are connected iff $w_{ij} = w_{ji} > 0$ ($w_{ij} = w_{ji} = 0$ otherwise). $W$ is a nonnegative matrix and doubly stochastic, i.e., $W\mathbf{1} = \mathbf{1}$ and $\mathbf{1}^T W = \mathbf{1}^T$. All diagonal elements $w_{ii}$ are strictly positive.*

**Assumption 1.2.** *(on the objective function) We assume the existence and the continuity of both $\nabla F_i(x)$ and $\nabla^2 F_i(x)$. Let $x^* \in \mathcal{K}$ denote the solution of the problem (4) such that $\mathcal{F}(x^*) = \min_{x \in \mathcal{K}} \mathcal{F}(x)$. We next assume that $\mathcal{F}(x)$ is $\lambda$-strongly convex where*

$$\mathcal{F}(y) \geq \mathcal{F}(x) + \langle \nabla \mathcal{F}(x), y - x \rangle + \frac{\lambda}{2}\|y - x\|^2, \ \forall x, y \in \mathcal{K}.$$

*We further assume the boundedness of the local Hessian where there exists a constant $c_1 \in \mathbb{R}^+$ such that*

$$\|\nabla^2 F_i(x)\|_2 \leq c_1, \ \forall x \in \mathcal{K}, \forall i \in \mathcal{N},$$

*where here it suffices to the spectral norm (keeping in mind for a matrix $A$, $\|A\|_2 \leq \|A\|_F$).*

**Assumption 1.3.** *(on the additive noise) $\zeta_{i,k}$ is a zero-mean uncorrelated noise with bounded variance, where $E(\zeta_{i,k}) = 0$ and $E(\zeta_{i,k}^2) = c_2 < \infty, \ \forall i \in \mathcal{N}$.*

**Lemma 1.4.** *(Qu & Li, 2018) Let $\rho_w$ be the spectral norm of $W - \frac{1}{n}\mathbf{1}\mathbf{1}^T$. When Assumption 1.1 is satisfied, we have the following inequality*

$$\|W\omega - \mathbf{1}\bar{\omega}\| \leq \rho_w\|\omega - \mathbf{1}\bar{\omega}\|, \ \forall \omega \in \mathbb{R}^{n \times d} \ and \ \bar{\omega} = \frac{1}{n}\mathbf{1}^T\omega,$$

*and $\rho_w < 1$.*

**Lemma 1.5.** *(Pu & Nedić, 2018) Define $h(\mathbf{x}) := \frac{1}{n}\mathbf{1}^T\nabla F(\mathbf{x}) \in \mathbb{R}^{1 \times d}$. Due to the boundedness of the second derivative in Assumption 1.2, there exists a scalar $L > 0$ such that the objective function is $L$-smooth, and*

$$\|\nabla \mathcal{F}(\bar{x}) - h(\mathbf{x})\| \leq \frac{L}{\sqrt{n}}\|\mathbf{x} - \mathbf{1}\bar{x}\|.$$

*Proof: See Appendix A.*

## 2 Distributed Stochastic Gradient Methods

We propose to employ a zero-order one-point estimate of the gradient subject to the stochastic process $S$ and an additive noise $\zeta$ while a stochastic perturbation and a step size are introduced, and we assume that

each agent can perform this estimation at each iteration. To elaborate, let $g_{i,k}$ denote the aforementioned gradient estimate for agent $i$ at time $k$, then we define it as

$$
\begin{aligned}
g_{i,k} &= \Phi_{i,k} \tilde{f}_i(x_{i,k} + \gamma_k \Phi_{i,k}, S_{i,k}) \\
&= \Phi_{i,k}(f_i(x_{i,k} + \gamma_k \Phi_{i,k}, S_{i,k}) + \zeta_{i,k}),
\end{aligned}
\tag{6}
$$

where $\gamma_k > 0$ is a vanishing step size and $\Phi_{i,k} \in \mathbb{R}^d$ is a perturbation randomly and independently generated by each agent $i$. $g_{i,k}$ is in fact a biased estimation of the gradient $\nabla F_i(x_{i,k})$ and the algorithm can converge under the condition that all parameters are properly chosen. For clarification on the form of this bias and more on the properties of this estimate, refer to Appendix B.

## 2.1 The 1P-DSG Algorithm

We consider a zero-order distributed stochastic gradient algorithm aiming for consensus with a one-point estimate. We denote it as 1P-DSG employing the gradient estimate $g_{i,k}$ in (6). Every agent $i$ initializes its variables with an arbitrary valued vector $x_{i,0} \in \mathcal{K}$ and computes $g_{i,0}$ at that variable. Then, at each time $k \in \mathbb{N}$, agent $i$ updates its variables independently according to the following steps:

$$
\begin{aligned}
z_{i,k+1} &= \sum_{j=1}^{n} w_{ij}(x_{j,k} - \alpha_k g_{j,k}) \\
x_{i,k+1} &= \Pi_{\mathcal{K}}(z_{i,k+1}) \\
&\text{perform the action: } x_{i,k+1} + \gamma_{k+1}\Phi_{i,k+1}
\end{aligned}
\tag{7}
$$

where $\alpha_k > 0$ is a step size. Algorithm (7) can then be written in the following compact matrix form for clarity of analysis:

$$
\begin{aligned}
\mathbf{z}_{k+1} &= W(\mathbf{x}_k - \alpha_k \mathbf{g}_k) \\
\mathbf{x}_{k+1} &= [x_{1,k+1}, x_{2,k+1}, \dots, x_{n,k+1}]^T \\
&\text{perform the action: } \mathbf{x}_{k+1} + \gamma_{k+1}\mathbf{\Phi}_{k+1}
\end{aligned}
\tag{8}
$$

where $\mathbf{\Phi}_k \in \mathbb{R}^{n \times d}$ is defined as $\mathbf{\Phi}_k = [\Phi_{1,k}, \Phi_{2,k}, \dots, \Phi_{n,k}]^T$.

As is evident from the update of the variables, the exchange between agents is limited to neighboring nodes, and it encompasses the value $\mathbf{x}_k - \alpha_k \mathbf{g}_k$ or the local gradient descent step.

We remark the effect of the gradient estimate variance on the convergence by carefully examining the steps in (8). Naturally, when the estimates have a large variance, the estimated gradients can vary widely from one sample to another. This means that the norm of $\mathbf{x}_{k+1}$, which is directly affected by this variance, may also grow considerably. Thus, it may then take longer to converge to the optimal solution because it cannot reliably discern the direction of the steepest descent. In the worst case, the huge variance causes instability as the optimizer may oscillate around the optimum or even diverge if the variance is too high, making converging to a satisfactory solution difficult. In this work, we use the fact that the local functions and the noise variance are bounded to prove that the variance of gradient estimate presented in (6) is indeed bounded. This boundedness, alongside the properties of the matrix $W$ in Assumption 1.1, allows us to find then an upper bound on the variation of $\mathbf{x}_{k+1}$ with respect to its mean and the variation of this mean with respect to the optimizer at every iteration and analyze the convergence of both.

We then consider the following assumptions for the subsequent convergence analysis. We must note that the first assumption is only taken into account when we study the algorithm's behavior with varying step sizes, otherwise it is dropped.

**Assumption 2.1.** *(on the step-sizes) Both $\alpha_k$ and $\gamma_k$ vanish to 0 as $k \to \infty$, and satisfy the the following sums*

$$
\sum_{k=1}^{\infty} \alpha_k \gamma_k = \infty, \ \sum_{k=1}^{\infty} \alpha_k^2 < \infty, \ \text{and} \ \sum_{k=1}^{\infty} \alpha_k \gamma_k^2 < \infty.
$$

**Assumption 2.2.** *(on the random perturbation) Let* $\Phi_{i,k} = (\phi_{i,k}^1, \phi_{i,k}^2, \ldots, \phi_{i,k}^d)^T$.

*Each agent $i$ chooses its $\Phi_{i,k}$ vector independently from other agents $j \neq i$. In addition, the elements of $\Phi_{i,k}$ are assumed i.i.d with $\mathbb{E}(\phi_{i,k}^{d_1}\phi_{i,k}^{d_2}) = 0$ for $d_1 \neq d_2$ and there exists $c_3 > 0$ such that $\mathbb{E}(\phi_{i,k}^{d_j})^2 = c_3$, $\forall d_j$, $\forall i$, almost surely. We further assume that there exists a constant $c_4 > 0$ where $\|\Phi_{i,k}\| \leq c_4$, $\forall i$, almost surely.*

**Example 2.3.** *One example is to take $\alpha_k = \alpha_0(k+1)^{-\upsilon_1}$ and $\gamma_k = \gamma_0(k+1)^{-\upsilon_2}$ with the constants $\alpha_0$, $\gamma_0$, $\upsilon_1$, $\upsilon_2 \in \mathbb{R}^+$. As $\sum_{k=1}^{\infty} \alpha_k\gamma_k$ diverges for $\upsilon_1 + \upsilon_2 \leq 1$, $\sum_{k=1}^{\infty} \alpha_k^2$ converges for $\upsilon_1 > 0.5$, and $\sum_{k=1}^{\infty} \alpha_k\gamma_k^2$ converges for $\upsilon_1 + 2\upsilon_2 > 1$, we can find pairs of $\upsilon_1$ and $\upsilon_2$ so that Assumption 2.1 is satisfied.*

*To achieve the conditions in Assumption 2.2, we can choose the probability distribution of $\phi_{i,k}^{d_j}$ to be the symmetrical Bernoulli distribution where $\phi_{i,k}^{d_j} \in \{-\frac{1}{\sqrt{d}}, \frac{1}{\sqrt{d}}\}$ with $\mathbb{P}(\phi_{i,k}^{d_j} = -\frac{1}{\sqrt{d}}) = \mathbb{P}(\phi_{i,k}^{d_j} = \frac{1}{\sqrt{d}}) = 0.5$, $\forall d_j$, $\forall i$.*

**Assumption 2.4.** *(on the local functions) $\mathcal{K}$ is a compact convex set and all local functions $x \mapsto f_i(x, S)$ are bounded on the $c_4\gamma_0$-neighborhood of $\mathcal{K}$, i.e.,*

$$|f_i(x, S)| < \infty, \quad \forall x \in N_{c_4\gamma_0}(\mathcal{K}), \forall S \in \mathbb{R}^m, \forall i \in \mathcal{N},$$

*where $N_{c_4\gamma_0}(\mathcal{K}) = \{x \in \mathbb{R}^d | \inf_{a \in \mathcal{K}} \|x - a\| < c_4\gamma_0\}$ is the $c_4\gamma_0$-neighborhood of $\mathcal{K}$.*

## 3 The 1P-DSG Algorithm

In this section, we analyze Algorithm 1P-DSG presented in (7) and (8).

### 3.1 Convergence Results

The goal of this part is to analyze the asymptotic behavior of Algorithm (8). We start the analysis by defining $\mathcal{H}_k$ as the history sequence $\{x_0, y_0, S_0, \ldots, x_{k-1}, y_{k-1}, S_{k-1}, x_k\}$ and denoting by $\mathbb{E}[.|\mathcal{H}_k]$ as the conditional expectation given $\mathcal{H}_k$.

We define $\tilde{g}_k$ to be the expected value of $\bar{g}_k$ with respect to all the stochastic terms $S, \Phi, \zeta$ given $\mathcal{H}_k$, i.e.,

$$\tilde{g}_k = \mathbb{E}_{S,\Phi,\zeta}[\bar{g}_k|\mathcal{H}_k].$$

In what follows, we use $\tilde{g}_k = \mathbb{E}[\bar{g}_k|\mathcal{H}_k]$ for shorthand notation.

We define the error $e_k$ to be the difference between the value of a single realization of $\bar{g}_k$ and its conditional expectation $\tilde{g}_k$, i.e.,

$$e_k = \bar{g}_k - \tilde{g}_k,$$

where $e_k$ can be seen as a stochastic noise. The following lemma describing the vanishing of the stochastic noise is essential for our main result.

**Lemma 3.1.** *If all Assumptions 1.2, 1.3, 2.1, 2.2, and 2.4 hold, then for any constant $\nu > 0$, we have*

$$\mathbb{P}(\lim_{K \to \infty} \sup_{K' \geq K} \| \sum_{k=K}^{K'} \alpha_k e_k \| \geq \nu) = 0, \ \forall \nu > 0.$$

*Proof: See Appendix C.*

For any integer $k \geq 0$, we define the divergence, or the error between the average action taken by the agents $\bar{x}_k$ and the optimal solution $x^*$ within $\mathcal{K}$ as

$$d_k = \|\bar{x}_k - x^*\|^2. \tag{9}$$

The following theorem describes the main convergence result.

**Theorem 3.2.** *If all Assumptions 1.1-1.3, 2.1-2.2, and 2.4 hold, then as $k \to \infty$, $d_k \to 0$, $\bar{x}_k \to x^*$, and $x_{i,k} \to \bar{x}_k$, for all $i \in \mathcal{N}$, almost surely by applying the Algorithm.*

*Proof: See Appendix D.*

## 3.2 Convergence Rate with Vanishing Step Sizes

This part deals with how fast the expected divergence vanishes to find the proposed algorithm's expected convergence rate. To do so, we define the expected divergence as

$$D_k = \mathbb{E}[\|\bar{x}_k - x^*\|^2].$$

The goal is to bound this divergence from above by sequences whose convergence rate is known. The analysis is highly associated with the parameters $\alpha_k$ and $\gamma_k$ that play a significant role in determining this upper bound. Hence, in what follows, the analysis starts with a general form of $\alpha_k$ and $\gamma_k$, then a particular case is considered.

### 3.2.1 General Form of $\alpha_k$ and $\gamma_k$

We first study the rate of convergence of the consensus error by introducing the following lemma.

**Lemma 3.3.** *Let Assumptions 1.1-1.3, 2.1-2.2, and 2.4 hold. Define*

$$K_1 = \underset{\frac{\alpha_{k+1}^2}{\alpha_k^2} > \frac{1+\rho_w^2}{2}}{\arg\min} \ k.$$

*Then, for $k \geq K_1$, there exist $0 < \vartheta_1, \vartheta_2 < \infty$, such that*

$$\|\mathbf{x}_k - \mathbf{1}\bar{x}_k\|^2 < \vartheta_1^2 \alpha_k^2 \ \ and \ \ \|\mathbf{z}_{k+1} - \mathbf{1}\bar{x}_k\|^2 \leq \vartheta_2^2 \alpha_k^2. \tag{10}$$

*Proof: Refer to Appendix D.3.*

Our main result regarding the convergence rate is summarized in the following theorem.

**Theorem 3.4.** *Let Assumptions 1.1-1.3, 2.1-2.2, and 2.4 hold. We then define the constants $A = \frac{\lambda c_3}{2}$, $B = \frac{4c_3 L^2 \vartheta_1^2}{\lambda n}$, $C = \frac{c_1^2 c_4^6}{c_3 \lambda}$, $E = \frac{\vartheta_2}{n}$,*

$$K_2 = \underset{A\alpha_k\gamma_k < 1}{\arg\min} k, \ \ and \ \ K_0 = \max\{K_1, K_2\}.$$

*We finally define the following parameters:*

$$\kappa_k = \frac{1 - (\frac{\gamma_{k+1}}{\gamma_k})^2}{\alpha_k \gamma_k}, \quad \sigma_1 = \max_{k \geq K_0} \kappa_k, \quad \sigma_2 = \max_{k \geq K_0} \frac{\alpha_k^2}{\gamma_k^2}, \quad \sigma_3 = \max_{k \geq K_0} \frac{\alpha_k}{\gamma_k^3},$$

$$\tau_k = \frac{1 - \frac{\alpha_{k+1}\gamma_{k+1}^{-1}}{\alpha_k\gamma_k^{-1}}}{\alpha_k\gamma_k}, \quad \sigma_4 = \max_{k \geq K_0} \tau_k, \quad \sigma_5 = \max_{k \geq K_0} \alpha_k\gamma_k, \quad \sigma_6 = \max_{k \geq K_0} \frac{\gamma_k^3}{\alpha_k}. \tag{11}$$

*If $\kappa_k < A$ for any $k \geq K_0$, then*

$$D_k \leq \varsigma_1 \gamma_k^2, \ \forall k \geq K_0, \tag{12}$$

*with*

$$\varsigma_1 \geq \max \left\{ \frac{D_{K_0}}{\gamma_{K_0}^2}, \frac{B\sigma_2 + E\sigma_3 + C}{A - \sigma_1} \right\}. \tag{13}$$

*If $\tau_k < A$ for any $k \geq K_0$, then*

$$D_k \leq \varsigma_2 \frac{\alpha_k}{\gamma_k}, \ \forall k \geq K_0, \tag{14}$$

*with*

$$\varsigma_2 \geq \max \left\{ \frac{D_{K_0}\gamma_{K_0}}{\alpha_{K_0}}, \frac{B\sigma_5 + C\sigma_6 + E}{A - \sigma_4} \right\}. \tag{15}$$

*Proof: See Appendix E.1.*

### 3.2.2 A Special Case of $\alpha_k$ and $\gamma_k$

We now consider the special case mentioned in Example 2.3:

$$\alpha_k = \alpha_0(k+1)^{-\upsilon_1} \text{ and } \gamma_k = \gamma_0(k+1)^{-\upsilon_2}, \tag{16}$$

where $0.5 < \upsilon_1 < 1$, $0 < \upsilon_2 \leq 1 - \upsilon_1$, and $\upsilon_1 + 2\upsilon_2 > 1$.

**Theorem 3.5.** *Let $\alpha_k$ and $\gamma_k$ have the forms given in (16) and consider the same assumptions of Theorem 3.4. If $\alpha_0\gamma_0 \geq \max\{2\upsilon_2, \upsilon_1 - \upsilon_2\}/A$, then we can say that there exists $\Upsilon < \infty$, where*

$$D_k \leq \Upsilon(k+1)^{-\min\{2\upsilon_2, \upsilon_1 - \upsilon_2\}}, \ \forall k \geq K_0.$$

*Proof: See Appendix E.4.*

The parameters clearly affect the upper bound of the convergence rate or rate of expected divergence decay in Theorem 3.5. As it is evident that

$$\max\{2\upsilon_2, \upsilon_1 - \upsilon_2\} \leq 0.5,$$

the best choice is when equality holds for $\upsilon_1 = 0.75$ and $\upsilon_2 = 0.25$. With the sufficient condition on the parameters in Theorem 3.5, we can finally state that our algorithm converges with a rate of $O(\frac{1}{\sqrt{k}})$ after a sufficient number of iterations $k > K_0$ when the step sizes are vanishing.

## 3.3 Regret Bound

To further examine the performance of our algorithm, we present the following theorem on the achieved regret bound.

**Theorem 3.6.** *Let the assumptions of Theorem 3.4 hold. When $\alpha_k$ and $\gamma_k$ have the forms of (16) with $\upsilon_1 = 0.75$ and $\upsilon_2 = 0.25$, the regret bound is given by*

$$\mathbb{E}\left[\frac{1}{n}\sum_{k=1}^{K}\sum_{i=1}^{n}F_i(x_{i,k}) - F_i(x^*)\right] \leq O(\sqrt{K}).$$

*Proof: See Appendix F.*

## 3.4 Convergence Rate with Constant Step Sizes

In this subsection, we fix the step sizes to $\alpha_k = \alpha > 0$ and $\gamma_k = \gamma > 0$, $\forall k \geq 0$, and we assume them to be two arbitrarily small values. We also define the following terms, $A = \frac{\lambda c_3}{2}$, $B = \frac{4c_3 L^2}{\lambda n}$, $C = \frac{c_1^2 c_4^6}{c_3 \lambda}$, and $R = \|\mathbf{x}_0 - \mathbf{1}\bar{x}_0\|^2$. We let $M$ denote the upper bound on $\|\bar{g}_k\|^2$. We then let $G_1 = \frac{2nM(1+\rho_w^2)}{(1-\rho_w^2)^2}$ and $G_2 = nM\left(\left(\frac{1+\rho_w^2}{1-\rho_w^2}\right)^2 + \frac{1+\rho_w^2}{1-\rho_w^2}\right)$. We finally define $\varrho_1 = 1 - A\alpha\gamma$ and $\varrho_2 = \frac{1+\rho_w^2}{2}$.

**Theorem 3.7.** *Assume $\alpha\gamma < \frac{1}{A}$ and $\alpha < \gamma$. Let Assumptions 1.1-1.3, 2.2, and 2.4 hold, then*

$$\|\mathbf{x}_{k+1} - \mathbf{1}\bar{x}_{k+1}\|^2 \leq \varrho_2^{k+1}R + \alpha^2 G_1 \ \text{ and } \ \|\mathbf{z}_{k+1} - \mathbf{1}\bar{x}_k\|^2 \leq \varrho_2^{k+1}R + \alpha^2 G_2. \tag{17}$$

*Meaning, $\|\mathbf{x}_{k+1} - \mathbf{1}\bar{x}_{k+1}\|^2$ converges with the linear rate of $O(\varrho_2^k)$ for an arbitrary small $\alpha$ almost surely. Further,*

- *When $\varrho_1 \leq \varrho_2$,*

$$D_{k+1} \leq \varrho_1^{k+1}D_0 + \varrho_2^{k+1}\frac{2R\left(B\alpha\gamma + \frac{\varrho_2}{n}\right)}{2A\alpha\gamma + \rho_w^2 - 1} + \alpha^2\frac{BG_1}{A} + \frac{\alpha}{\gamma}\frac{G_2}{nA} + \gamma^2\frac{C}{A}. \tag{18}$$

*Then, for arbitrary small step sizes, $D_k$ converges with the linear rate of $O(\varrho_2^k)$.*

- *When $\varrho_1 > \varrho_2$,*

$$D_{k+1} \leq \varrho_1^{k+1}\Big(D_0 + \frac{2RB\alpha\gamma + \frac{2R\varrho_2}{n}}{1 - 2A\alpha\gamma - \rho_w^2}\Big) + \alpha^2\frac{BG_1}{A} + \frac{\alpha}{\gamma}\frac{G_2}{nA} + \gamma^2\frac{C}{A}. \tag{19}$$

*Then, for arbitrary small step sizes, $D_k$ converges with the linear rate of $O\big(\varrho_1^k\big)$.*

*Proof: See Appendix G.*

Taking arbitrarily small values of $\alpha$, $\gamma$ satisfying $\alpha\gamma < \frac{1}{A}$ and $\alpha < \gamma$, and setting $\varrho = \max\{\varrho_1, \varrho_2\}$, the convergence rate becomes $O(\varrho^k)$, achieving the same rate as with FO information.

## 4 Numerical Results

In this section, we provide numerical examples to illustrate the performance of the algorithm 1P-DSG. We compare it with FO distributed methods aiming to achieve consensus, namely DSGT (Pu & Nedić, 2018) and EXTRA (Shi et al., 2015), a ZO distributed algorithm denoted 2P-DSG based on the two-point estimate in (2) (Tang et al., 2021), and a ZO centralized algorithm based on gradient descent (e.g. Flaxman et al. (2004) and Bach & Perchet (2016)) using another one-point estimate which is presented in (1). We denote the ZO centralized algorithm by 1P-GD. We also compare with a centralized version of our algorithm where we use the estimate in (6). For DSGT and EXTRA, we calculate the exact gradient and add white noise to it to form an unbiased FO estimator and for all the ZO algorithms, we consider that the function queries are noisy. The network topology is a connected Erdős-Rényi random graph with a probability of 0.05.

We consider a logistic classification problem to classify $m$ images of the two digits, labeled as $y_{ij} = +1$ or $-1$ from the MNIST data set (LeCun & Cortes, 2005). Each image, $X_{ij}$, is a 785-dimensional vector and is compressed using a lossy autoencoder to become 10-dimensional denoted as $X'_{ij}$, i.e., $d = 10$. The total images are split equally among the agents such that each agent has $m_i = \frac{m}{n}$ images and no access to other ones for privacy constraints. However, the goal is still to make use of all images and to solve collaboratively

$$\min_{\theta \in \mathcal{K}} \frac{1}{n}\sum_{i=1}^n \frac{1}{m}\sum_{j=1}^{m_i} \mathbb{E}_{u\sim\mathcal{N}(1,\sigma_u)} \ln(1 + \exp(-u_{ij}y_{ij}.X'^T_{ij}\theta)) + c\|\theta\|^2,$$

while reaching consensus on the decision variable $\theta \in \mathcal{K}$ with $\mathcal{K} = [-10, 10]^d$. We note here that $u$ models some perturbation on the local querying of every example to add to the randomization of the communication process.

We consider classifying the digits 1 and 2 with $m = 12700$ images. There are $n = 100$ agents in the network and thus each has a local batch of $m_i = 127$ images. We take $\sigma_u = 0.01$ and let $\alpha_k = 0.05(k + 1)^{-0.75}$ and $\gamma_k = 0.8(k + 1)^{-0.25}$ for 1P-DSG with vanishing step sizes, and $\alpha = 0.05$ and $\gamma = 0.6$ with constant step sizes. We choose $\Phi_k \in \{-\frac{1}{\sqrt{d}}, \frac{1}{\sqrt{d}}\}^d$ with equal probability. Also, every function query is subject to a white noise generated by the standard normal distribution. For the DSGT algorithm, we set the step size to $\alpha_k = 0.015(k + 1)^{-1}$ when it is vanishing and $\alpha = 0.015$ when constant, and we do not consider the perturbation on the objective function nor the noise on the objective function, only the noise on the exact gradient. Similarly for EXTRA and we set its step size to $\alpha = 0.01$. For 2P-DSG, we consider $\alpha_k = 0.01(k + 1)^{-0.75}$ and $\gamma_k = 0.01(k + 1)^{-0.25}$. For the centralized 1P-GD algorithm, we set $\alpha = 0.005$ and $\gamma = 0.5$ ($\alpha = 0.03$ and $\gamma = 0.6$ with estimator (6)). We let $c = 0.1$, and the initialization be the same for all algorithms, with $\theta_{i,0}$ uniformly chosen from $[-0.5, 0.5]^d$, $\forall i \in \mathcal{N}$, per instance. We finally average the simulations over 30 instances.

The expected evolution of the loss objective function is presented in Figure 1 and the graphs are zoomed in on in Figure 2. Experimental results seem to validate our theoretical results: Our proposed algorithm converges linearly fast with constant step sizes, however the final gap is due to converging to an $O(\alpha)$-neighborhood of the optimal solution. 1P-DSG with vanishing step sizes converges with an $O(\frac{1}{\sqrt{k}})$ while DSGT with vanishing step size converges at a rate of $O(\frac{1}{k})$. Using constant vs vanishing step size does not

seem to affect the convergence rate of the loss function of DSGT. EXTRA consistently performs similarly to DSGT. The most interesting point is that 1P-DSG, with vanishing and constant step sizes, outperforms the centralized ZO counterpart 1P-GD highlighting an advantage of our gradient estimate. In addition, there seems to be an evident advantage to using our one-point estimate to two-point one when the queries are noisy, as our algorithm outperforms 2P-DSG. We also note that the estimate we use seem more stable than the other ZO counterparts as shown in Figure 3 where we plot the average loss error bar of these algorithms. A possible explanation is that not dividing by $\gamma$ in the estimate provides some stability against noisy queries (this is slightly evident in 1P-GD with the different estimators and in 1P-DSG vs. 2P-DSG).

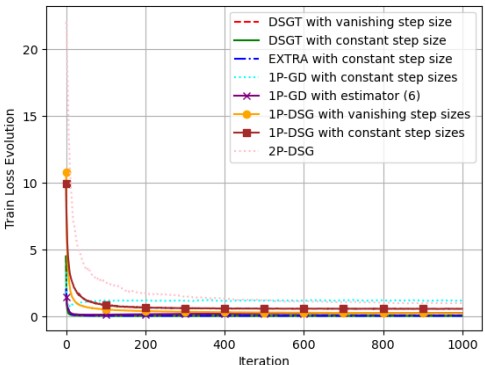

Figure 1: Expected loss function evolution of the proposed algorithm vs. DSGT, EXTRA, and 1P-GD considering vanishing vs. constant step sizes.

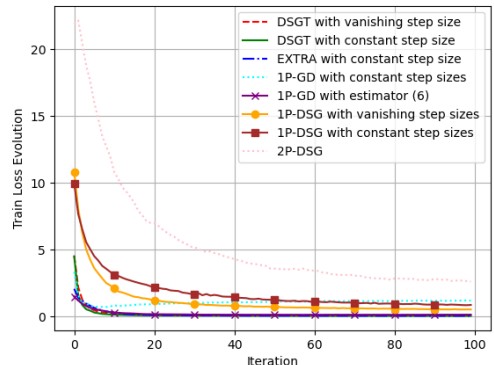

Figure 2: Expected loss function evolution of the proposed algorithm vs. DSGT, EXTRA, and 1P-GD considering vanishing vs. constant step sizes.

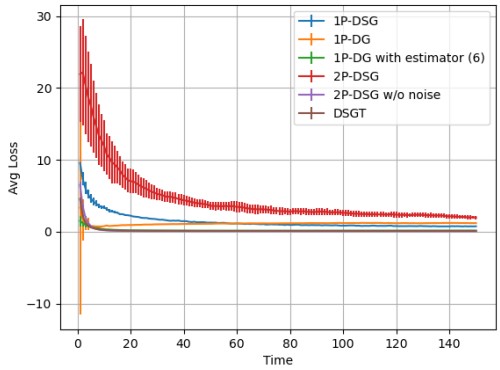

Figure 3: The average loss error bar evolution of the proposed algorithm vs. DSGT.

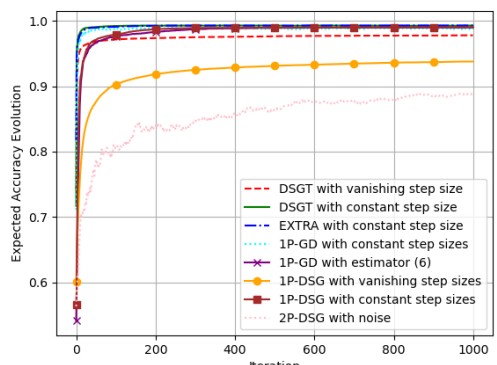

Figure 4: Expected test accuracy evolution of the proposed algorithm vs. DSGT, EXTRA, and 1P-GD considering vanishing vs. constant step sizes.

In Figure 4, we measure at every iteration the classification accuracy against an independent test set of 2167 images using the updated mean vector $\bar{\theta}_k = \frac{1}{n}\sum_{i=1}^{n}\theta_{i,k}$ of the local decision variables. The interest of the constant step sizes appears in the convergence rate of this accuracy, where our algorithm is able to compete with DSGT with full FO information, and to outperform DSGT with a vanishing step size. This is an important result as it shows that the classification goal with ZO is well met despite the limiting upper bounds of convergence rate and that $O(\alpha)$-neighborhood of the optimal solution achieved linearly fast can be sufficient to achieve the best possible accuracy.

The reason for this better accuracy attainment is generally because the step sizes affect the bound on the generalization error. For example, Hardt et al. (2016) prove theoretically that the bound on the generalization error for strongly convex objectives is smaller when the step sizes are constant (theorem 3.9) than when they

are vanishing (theorem 3.10), wherein the latter, there is an extra element containing the supremum of the function. Naturally, the step sizes seem to play a role in affecting the bound on the evolution of iterates, which in turn affects the uniform stability of the SGD method (stable means that the loss function is not affected much if one datapoint is different) and the generalization error of an SGD-trained model is upper bounded by the uniform stability bound.

This result seems to be confirmed by the centralized 1P-GD vs 1P-DSG with vanishing step sizes. Despite the latter outperforming the first in convergence speed (of the objective function), the first with constant step sizes seems to generalize better.

In Figures 5, 6, and 7 the curves are those of the evolution of the expected consensus error, or $\mathbb{E}\left[\sum_{i=1}^{n}\|\theta_{i,k} - \bar{\theta}_k\|^2\right]$ which is the expected error between the local decision variables and their average. For all algorithms, the error again validates the theoretical bounds and decreases quite fast. Generally, as evident in Figure 7 for all algorithms (expect 2P-DSG where a noise term is always multiplied by $\frac{1}{\gamma_k}$), vanishing step sizes allow the consensus error to completely vanish while constant step sizes leave an $O(\alpha^2)$-gap.

We add other numerical examples for different image labels in Appendix H.

## 5   Conclusion

In this work, we extended the distributed stochastic gradient algorithm to present a practical solution to a relevant problem with realistic assumptions. A novel ZO algorithm was studied and proved to converge with a biased and high variance one-point gradient estimate and a stochastic perturbation on the objective function. In the context of noisy ZO optimization, we have successfully established a linear convergence rate of $O(\varrho^k)$ using fixed step sizes and $O(\frac{1}{\sqrt{k}})$ with vanishing step sizes. These rates align with the optimal expectations examined in the existing literature. We also prove a regret bound that of $O(\sqrt{k})$ with vanishing step sizes. A numerical application confirmed the success and efficiency of the algorithm.

## 6   Acknowledgment

We are grateful for the comments made by the editor and the reviewers that have substantially improved the quality of this work.

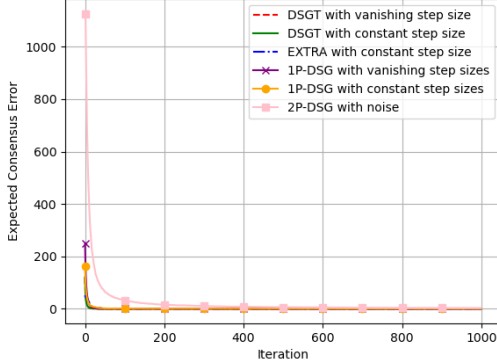

Figure 5: Expected consensus error evolution of the proposed algorithm vs. DSGT and EXTRA considering vanishing vs. constant step sizes.

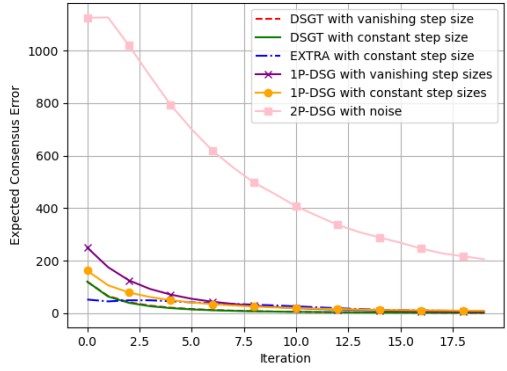

Figure 6: Expected consensus error evolution of the proposed algorithm vs. DSGT and EXTRA considering vanishing vs. constant step sizes.

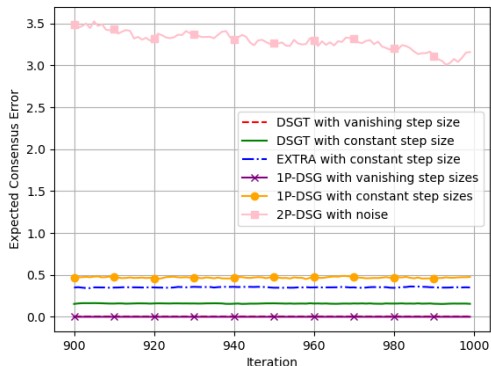

Figure 7: Expected consensus error evolution of the proposed algorithm vs. DSGT and EXTRA considering vanishing vs. constant step sizes.

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

## A   L-Smoothness Property

$$
\begin{aligned}
\|\nabla\mathcal{F}(\bar{x}) - h(\mathbf{x})\| =& \left\|\frac{1}{n}\sum_{i=1}^{n}\Big(\nabla F_i(\bar{x}) - \nabla F_i(x_i)\Big)\right\| \\
\leq& \frac{1}{n}\sum_{i=1}^{n}\left\|\nabla F_i(\bar{x}) - \nabla F_i(x_i)\right\| \\
\leq& \frac{L}{n}\sum_{i=1}^{n}\|\bar{x} - x_i\| \\
=& \frac{L}{n}\sum_{i=1}^{n}\|x_i - \bar{x}\| \\
\overset{(a)}{\leq}& \frac{L}{n}\sqrt{n\sum_{i=1}^{n}\|x_i - \bar{x}\|^2} \\
\overset{(b)}{=}& \frac{L}{\sqrt{n}}\|\mathbf{x} - \mathbf{1}\bar{x}\|,
\end{aligned}
$$

where $(a)$ is by applying the Cauchy-Schwarz inequality, $|\sum_{i=1}^{n} a_i \cdot 1| \leq (\sum_{i=1}^{n} a_i^2)^{\frac{1}{2}} \cdot (\sum_{i=1}^{n} 1^2)^{\frac{1}{2}} = n^{\frac{1}{2}}(\sum_{i=1}^{n} a_i^2)^{\frac{1}{2}}$, and $(b)$ is by definition of the Frobenius norm, $\|\mathbf{x} - \mathbf{1}\bar{x}\|^2 = \sum_{i=1}^{n}\|x_i - \bar{x}\|^2$.

## B   Estimated Gradient

In this section, we derive the bias of the gradient estimate with respect to the real gradient of the local objective function. Let

$$
\breve{g}_{i,k} = \mathbb{E}_{S,\Phi,\zeta}[g_{i,k}|\mathcal{H}_k].
$$

Thus, by Assumption 1.3 and the definition in (4),

$$
\begin{aligned}
\breve{g}_{i,k} &= \mathbb{E}_{S,\Phi,\zeta}[\Phi_{i,k}(f_i(x_{i,k} + \gamma_k\Phi_{i,k}, S_{i,k}) + \zeta_{i,k})|\mathcal{H}_k] \\
&= \mathbb{E}_{S,\Phi}[\Phi_{i,k}f_i(x_{i,k} + \gamma_k\Phi_{i,k}, S_{i,k})|\mathcal{H}_k] \\
&= \mathbb{E}_{\Phi}[\Phi_{i,k}F_i(x_{i,k} + \gamma_k\Phi_{i,k})|\mathcal{H}_k].
\end{aligned}
$$

By Taylor's theorem and the mean-valued theorem, there exists $\tilde{x}_{i,k}$ located between $x_{i,k}$ and $x_{i,k} + \gamma_k \Phi_{i,k}$ where

$$F_i(x_{i,k} + \gamma_k \Phi_{i,k}) = F_i(x_{i,k}) + \gamma_k \langle \Phi_{i,k}, \nabla F_i(x_{i,k}) \rangle + \frac{\gamma_k^2}{2} \langle \Phi_{i,k}, \nabla^2 F_i(\tilde{x}_{i,k}) \Phi_{i,k} \rangle,$$

substituting in the previous definition,

$$\begin{aligned}
\breve{g}_{i,k} &= F_i(x_{i,k}) \mathbb{E}_\Phi[\Phi_{i,k}] + \gamma_k \mathbb{E}_\Phi[\Phi_{i,k} \Phi_{i,k}^T] \nabla F_i(x_{i,k}) + \frac{\gamma_k^2}{2} \mathbb{E}_\Phi[\Phi_{i,k} \Phi_{i,k}^T \nabla^2 F_i(\tilde{x}_{i,k}) \Phi_{i,k} | \mathcal{H}_k] \\
&= c_3 \gamma_k [\nabla F_i(x_{i,k}) + b_{i,k}].
\end{aligned}$$

Thus, the estimation bias has the form

$$\begin{aligned}
b_{i,k} &= \frac{\breve{g}_{i,k}}{c_3 \gamma_k} - \nabla F_i(x_{i,k}) \\
&= \frac{\gamma_k}{2c_3} \mathbb{E}_\Phi[\Phi_{i,k} \Phi_{i,k}^T \nabla^2 F_i(\tilde{x}_{i,k}) \Phi_{i,k} | \mathcal{H}_k].
\end{aligned}$$

Let Assumptions 1.2 and 2.2 hold. Then, we can bound the bias as

$$\begin{aligned}
\|b_{i,k}\| &\leq \frac{\gamma_k}{2c_3} \mathbb{E}_\Phi[\|\Phi_{i,k}\|_2 \|\Phi_{i,k}^T\|_2 \|\nabla^2 F_i(\tilde{x}_{i,k})\|_2 \|\Phi_{i,k}\|_2 | \mathcal{H}_k] \\
&\leq \gamma_k \frac{c_4^3 c_1}{2c_3}.
\end{aligned} \tag{20}$$

We can see $\|b_{i,k}\| \to 0$ as $k \to \infty$ since $\gamma_k$ is vanishing. We remark that

$$\begin{aligned}
\tilde{g}_k &= \mathbb{E}[\bar{g}_k | \mathcal{H}_k] \\
&= \frac{1}{n} \sum_{i=1}^n \mathbb{E}[g_{i,k} | \mathcal{H}_k] \\
&= \frac{1}{n} \sum_{i=1}^n c_3 \gamma_k [\nabla F_i(x_{i,k}) + b_{i,k}] \\
&= c_3 \gamma_k [h(\mathbf{x}_k) + \bar{b}_k]
\end{aligned} \tag{21}$$

is also a biased estimator of $h(\mathbf{x}_k)$ with

$$\begin{aligned}
\|\bar{b}_k\| &= \|\frac{1}{n} \sum_{i=1}^n b_{i,k}\| \\
&\leq \frac{1}{n} \sum_{i=1}^n \|b_{i,k}\| \\
&\leq \frac{1}{n} \sum_{i=1}^n \gamma_k \frac{c_4^3 c_1}{2c_3} \\
&= \gamma_k \frac{c_4^3 c_1}{2c_3}.
\end{aligned} \tag{22}$$

**Lemma B.1.** *Let all Assumptions 1.3, 2.2, and 2.4 hold, then there exists a bounded constant $\bar{M} > 0$, such that $\mathbb{E}[\|\bar{g}_k\|^2 | \mathcal{H}_k] < \bar{M}$.*

*Proof.* $\forall i \in \mathcal{N}$, we have

$$\begin{aligned}
\mathbb{E}[\|g_{i,k}\|^2 | \mathcal{H}_k] &= \mathbb{E}[\|\Phi_{i,k}(f_i(x_{i,k} + \gamma_k \Phi_{i,k}, S_{i,k}) + \zeta_{i,k})\|^2 | \mathcal{H}_k] \\
&= \mathbb{E}[\|\Phi_{i,k}\|^2 \|f_i(x_{i,k} + \gamma_k \Phi_{i,k}, S_{i,k}) + \zeta_{i,k}\|^2 | \mathcal{H}_k] \\
&\overset{(a)}{\leq} c_4^2 \mathbb{E}[(f_i(x_{i,k} + \gamma_k \Phi_{i,k}, S_{i,k}) + \zeta_{i,k})^2 | \mathcal{H}_k] \\
&\overset{(b)}{=} c_4^2 \mathbb{E}[f_i^2(x_{i,k} + \gamma_k \Phi_{i,k}, S_{i,k}) | \mathcal{H}_k] + c_4^2 c_2 \\
&\overset{(f)}{<} \infty,
\end{aligned}$$

where $(a)$ is due to Assumption 2.2, $(b)$ Assumption 1.3, and $(c)$ Assumption 2.4.

Then, $\mathbb{E}[\|\mathbf{g}_k\|^2|\mathcal{H}_k] = \mathbb{E}\left[\sum_{i=1}^n \|g_{i,k}\|^2\Big|\mathcal{H}_k\right] = \sum_{i=1}^n \mathbb{E}[\|g_{i,k}\|^2|\mathcal{H}_k] < \infty$ and

$$
\begin{aligned}
\mathbb{E}[\|\bar{g}_k\|^2|\mathcal{H}_k] =& \mathbb{E}\left[\|\frac{1}{n}\sum_{i=1}^n g_{i,k}\|^2\Big|\mathcal{H}_k\right] \\
=& \frac{1}{n^2}\mathbb{E}\left[\|\sum_{i=1}^n g_{i,k}\|^2\Big|\mathcal{H}_k\right] \\
\leq& \frac{n}{n^2}\mathbb{E}\left[\sum_{i=1}^n \|g_{i,k}\|^2\Big|\mathcal{H}_k\right] \\
=& \frac{1}{n}\sum_{i=1}^n \mathbb{E}[\|g_{i,k}\|^2|\mathcal{H}_k] \\
<& \infty.
\end{aligned}
$$

$\square$

## C  Stochastic Noise

To prove Lemma 3.1, we begin by demonstrating that the sequence $\{\sum_{k=K}^{K'} \alpha_k e_k\}_{K' \geq K}$ is a martingale. To do so, we have to prove that for all $K' \geq K$, $X_{K'} = \sum_{k=K}^{K'} \alpha_k e_k$ satisfies the following two conditions:

(i) $\mathbb{E}[X_{K'+1}|X_{K'}] = X_{K'}$

(ii) $\mathbb{E}[\|X_{K'}\|^2] < \infty$

We know that

$$
\mathbb{E}[e_k] = \mathbb{E}[\bar{g}_k - \mathbb{E}[\bar{g}_k|\mathcal{H}_k]] = \mathbb{E}_{\mathcal{H}_k}\left[\mathbb{E}\left[\bar{g}_k - \mathbb{E}[\bar{g}_k|\mathcal{H}_k]\Big|\mathcal{H}_k\right]\right] = 0
$$

by the law of total expectation. Hence,

$$
\mathbb{E}[X_{K'+1}|X_{K'}] = \mathbb{E}\left[\alpha_{K'+1}e_{K'+1} + \sum_{k=K}^{K'}\alpha_k e_k\Big|\sum_{k=K}^{K'}\alpha_k e_k\right] = 0 + \sum_{k=K}^{K'}\alpha_k e_k = X_{K'}. \tag{23}
$$

In addition, $e_k$ and $e_{k'}$ are uncorrelated for any $k \neq k'$ since (assuming $k > k'$) $\mathbb{E}[e_k^T e_{k'}] = \mathbb{E}[\mathbb{E}[e_k^T e_{k'}|\mathcal{H}_k]] = \mathbb{E}[e_{k'}\mathbb{E}[e_k^T|\mathcal{H}_k]] = 0$. Thus,

$$
\begin{aligned}
\mathbb{E}(\|\sum_{k=K}^{K'}\alpha_k e_k\|^2) =& \mathbb{E}(\sum_{k=K}^{K'}\sum_{k'=K}^{K'}\alpha_k\alpha_{k'}\langle e_k, e_{k'}\rangle) \\
\overset{(a)}{=}& \mathbb{E}(\sum_{k=K}^{K'}\|\alpha_k e_k\|^2) \\
\leq& \sum_{k=K}^{\infty}\mathbb{E}(\alpha_k^2\|\bar{g}_k - \mathbb{E}[\bar{g}_k|\mathcal{H}_k]\|^2) \\
=& \sum_{k=K}^{\infty}\alpha_k^2\mathbb{E}(\|\bar{g}_k\|^2) - \mathbb{E}_{\mathcal{H}_k}(\|\mathbb{E}[\bar{g}_k|\mathcal{H}_k]\|^2) \\
\leq& \sum_{k=K}^{\infty}\alpha_k^2\mathbb{E}(\|\bar{g}_k\|^2) \\
\overset{(b)}{\leq}& M\sum_{k=K}^{\infty}\alpha_k^2 \overset{(c)}{<} \infty,
\end{aligned} \tag{24}
$$

where $(a)$ is due to the uncorrelatedness $\mathbb{E}[\langle e_k, e_{k'} \rangle] = 0$, $(b)$ is by Lemma B.1, and $(c)$ is by Assumption 2.1. Therefore, both (i) and (ii) are satisfied and we can say that $\{\sum_{k=K}^{K'} \alpha_k e_k\}_{K' \geq K}$ is a martingale. This permits us to use Doob's martingale inequality Doob (1953):

For any constant $\nu > 0$,

$$
\begin{aligned}
\mathbb{P}(\sup_{K' \geq K} \|\sum_{k=K}^{K'} \alpha_k e_k\| \geq \nu) &\leq \frac{1}{\nu^2} \mathbb{E}(\|\sum_{k=K}^{K'} \alpha_k e_k\|^2) \\
&\overset{(a)}{\leq} \frac{M}{\nu^2} \sum_{k=K}^{\infty} \alpha_k^2,
\end{aligned}
\tag{25}
$$

where $(a)$ is following the exact same steps as (24).

Since $M$ is a bounded constant and $\lim_{K \to \infty} \sum_{k=K}^{\infty} \alpha_k^2 = 0$ by Assumption 2.1, we get $\lim_{K \to \infty} \frac{M}{\nu^2} \sum_{k=K}^{\infty} \alpha_k^2 = 0$ for any bounded constant $\nu$. Hence, the probability that $\|\sum_{k=K}^{K'} \alpha_k e_k\| \geq \nu$ also vanishes as $K \to \infty$, which concludes the proof.

# D  Proof of Convergence

We start by stating the following lemma that will be useful for the proof of convergence.

**Lemma D.1.** *If all Assumptions 1.1-1.3, 2.1-2.2, and 2.4 hold, then $\lim_{k \to \infty} \|\mathbf{x}_k - \mathbf{1}\bar{x}_k\|^2 = 0$. In fact, we have*

$$
\sum_{k=0}^{\infty} \|\mathbf{x}_k - \mathbf{1}\bar{x}_k\|^2 < \infty, \; \sum_{k=0}^{\infty} \|\mathbf{z}_{k+1} - \mathbf{1}\bar{x}_k\|^2 < \infty, \; and \; \sum_{k=0}^{\infty} \gamma_k \alpha_k \|\mathbf{x}_k - \mathbf{1}\bar{x}_k\| < \infty,
$$

*almost surely.*

*Proof: See Appendix D.2.*

## D.1  Proof of Theorem 3.2

By using the compact form of the algorithm in (8), we know that

$$
\bar{z}_{k+1} = \frac{1}{n}\mathbf{1}^T W(\mathbf{x}_k - \alpha_k \mathbf{g}_k) \overset{(a)}{=} \frac{1}{n}\mathbf{1}^T(\mathbf{x}_k - \alpha_k \mathbf{g}_k) = \bar{x}_k - \alpha_k \bar{g}_k,
\tag{26}
$$

where $(a)$ is again due to the doubly stochastic property of $W$.

The divergence at time $k+1$ can then be written as

$$
\begin{aligned}
d_{k+1} &= \|\bar{x}_{k+1} - x^*\|^2 \\
&= \|\frac{1}{n}\sum_{i=1}^{n}(x_{i,k+1} - x^*)\|^2 \\
&\leq \frac{n}{n^2} \sum_{i=1}^{n} \|x_{i,k+1} - x^*\|^2 \\
&\overset{(a)}{\leq} \frac{1}{n} \sum_{i=1}^{n} \|z_{i,k+1} - x^*\|^2
\end{aligned}
$$

$$
\begin{aligned}
&= \frac{1}{n} \sum_{i=1}^{n} \| z_{i,k+1} - \bar{x}_k + \bar{x}_k - x^* \|^2 \\
&= \frac{1}{n} \sum_{i=1}^{n} \| z_{i,k+1} - \bar{x}_k \|^2 + 2 \frac{1}{n} \sum_{i=1}^{n} \langle z_{i,k+1} - \bar{x}_k, \bar{x}_k - x^* \rangle + \frac{1}{n} \sum_{i=1}^{n} \| \bar{x}_k - x^* \|^2 \\
&= \frac{1}{n} \| \mathbf{z}_{k+1} - \mathbf{1}\bar{x}_k \|^2 + 2 \langle \bar{z}_{k+1} - \bar{x}_k, \bar{x}_k - x^* \rangle + \| \bar{x}_k - x^* \|^2 \\
&\overset{(b)}{=} \frac{1}{n} \| \mathbf{z}_{k+1} - \mathbf{1}\bar{x}_k \|^2 + 2 \langle -\alpha_k \bar{g}_k, \bar{x}_k - x^* \rangle + d_k \\
&= d_k - 2\alpha_k \langle \bar{x}_k - x^*, \bar{g}_k - \mathbb{E}[\bar{g}_k | \mathcal{H}_k] + \mathbb{E}[\bar{g}_k | \mathcal{H}_k] \rangle + \frac{1}{n} \| \mathbf{z}_{k+1} - \mathbf{1}\bar{x}_k \|^2 \\
&= d_k - 2\alpha_k \langle \bar{x}_k - x^*, \mathbb{E}[\bar{g}_k | \mathcal{H}_k] \rangle - 2\alpha_k \langle \bar{x}_k - x^*, e_k \rangle + \frac{1}{n} \| \mathbf{z}_{k+1} - \mathbf{1}\bar{x}_k \|^2 \\
&\overset{(c)}{=} d_k - 2c_3 \gamma_k \alpha_k \langle \bar{x}_k - x^*, h(\mathbf{x}_k) + \bar{b}_k \rangle - 2\alpha_k \langle \bar{x}_k - x^*, e_k \rangle + \frac{1}{n} \| \mathbf{z}_{k+1} - \mathbf{1}\bar{x}_k \|^2 \\
&= d_k - 2c_3 \gamma_k \alpha_k \langle \bar{x}_k - x^*, \nabla \mathcal{F}(\bar{x}_k) \rangle + 2c_3 \gamma_k \alpha_k \langle \bar{x}_k - x^*, \nabla \mathcal{F}(\bar{x}_k) - h(\mathbf{x}_k) \rangle \\
&\quad - 2c_3 \gamma_k \alpha_k \langle \bar{x}_k - x^*, \bar{b}_k \rangle - 2\alpha_k \langle \bar{x}_k - x^*, e_k \rangle + \frac{1}{n} \| \mathbf{z}_{k+1} - \mathbf{1}\bar{x}_k \|^2 \\
&\overset{(d)}{\leq} d_k - 2c_3 \gamma_k \alpha_k \langle \bar{x}_k - x^*, \nabla \mathcal{F}(\bar{x}_k) \rangle + \frac{2c_3 L \gamma_k \alpha_k}{\sqrt{n}} \| \bar{x}_k - x^* \| \| \mathbf{x}_k - \mathbf{1}\bar{x}_k \| \\
&\quad + 2c_3 \gamma_k \alpha_k \| \bar{x}_k - x^* \| \| \bar{b}_k \| - 2\alpha_k \langle \bar{x}_k - x^*, e_k \rangle + \frac{1}{n} \| \mathbf{z}_{k+1} - \mathbf{1}\bar{x}_k \|^2,
\end{aligned}
\tag{27}
$$

where $(a)$ is by the projection inequality (5) noting that $x^* \in \mathcal{K}$ (so projecting it onto $\mathcal{K}$ gives us the same point), $(b)$ is by (26), $(c)$ is due to (21), and $(d)$ is due to Lemma 1.5.

By recursion of inequality (27), we have

$$
\begin{aligned}
d_{K+1} \leq &\ d_0 - 2c_3 \sum_{k=0}^{K} \gamma_k \alpha_k \langle \bar{x}_k - x^*, \nabla \mathcal{F}(\bar{x}_k) + \bar{b}_k \rangle + \frac{2c_3 L}{\sqrt{n}} \sum_{k=0}^{K} \gamma_k \alpha_k \| \bar{x}_k - x^* \| \| \mathbf{x}_k - \mathbf{1}\bar{x}_k \| \\
&+ 2c_3 \sum_{k=0}^{K} \gamma_k \alpha_k \| \bar{x}_k - x^* \| \| \bar{b}_k \| - 2 \sum_{k=0}^{K} \alpha_k \langle \bar{x}_k - x^*, e_k \rangle + \frac{1}{n} \sum_{k=0}^{K} \| \mathbf{z}_{k+1} - \mathbf{1}\bar{x}_k \|^2.
\end{aligned}
\tag{28}
$$

By Lemma 3.1, we have $\lim_{K \to \infty} \| \sum_{k=0}^{K} \alpha_k e_k \| < \infty$ almost surely. Since $\| \bar{x}_k - x^* \| < \infty$ by the compactness of $\mathcal{K}$ in Assumption 2.4, hence

$$
\lim_{K \to \infty} \| \sum_{k=0}^{K} \alpha_k \langle \bar{x}_k - x^*, e_k \rangle \| < \infty.
\tag{29}
$$

From (42) in Lemma D.1, we have

$$
\lim_{K \to \infty} \sum_{k=0}^{K} \| \mathbf{z}_{k+1} - \mathbf{1}\bar{x}_k \|^2 < \infty.
\tag{30}
$$

As stated in Lemma D.1, we have $\sum_{k=0}^{\infty} \gamma_k \alpha_k \| \mathbf{x}_k - \mathbf{1}\bar{x}_k \| < \infty$, adding to $\| \bar{x}_k - x^* \| < \infty$ by Assumption 2.4, then

$$
\lim_{K \to \infty} \sum_{k=0}^{K} \gamma_k \alpha_k \| \bar{x}_k - x^* \| \| \mathbf{x}_k - \mathbf{1}\bar{x}_k \| < \infty.
\tag{31}
$$

By (22), we know that $\| \bar{b}_k \| \leq \frac{c_4^3 c_1}{2c_3} \gamma_k$ and $\| \bar{x}_k - x^* \| < \infty$ by Assumption 2.4,

$$
\lim_{K \to \infty} \sum_{k=0}^{K} \gamma_k^2 \alpha_k \| \bar{x}_k - x^* \| < \infty,
\tag{32}
$$

by Assumption 2.1.

From the above inequalities (28)-(32), we see that there exists $0 < D' < \infty$ such that $d_{K+1} \leq D' + z_K$, with $z_K$ defined as

$$z_K = -2c_3 \sum_{k=0}^{K} \gamma_k \alpha_k \langle \bar{x}_k - x^*, \nabla \mathcal{F}(\bar{x}_k) \rangle. \tag{33}$$

By the strong convexity, we have

$$- \langle \bar{x}_k - x^*, \nabla \mathcal{F}(\bar{x}_k) \rangle \leq \mathcal{F}(x^*) - \mathcal{F}(\bar{x}_k) - \frac{\lambda}{2} \|\bar{x}_k - x^*\|^2 \leq 0, \tag{34}$$

as $\mathcal{F}(\bar{x}_k) \geq \mathcal{F}(x^*)$ by the definition of $x^*$ being the optimum in $\mathcal{K}$ and $\bar{x}_k \in \mathcal{K}$ (by the property of a convex set).

Thus, $z_K \leq 0$, confirming $d_{K+1} < \infty$.

Let's assume that $\forall \epsilon_h > 0, \exists K_h$ such that $\|\bar{x}_k - x^*\|^2 > \epsilon_h$ for $k \geq K_h$, meaning

$$\lim_{K \to \infty} - \sum_{k=K_h}^{K} \gamma_k \alpha_k \|\bar{x}_k - x^*\|^2 < -\epsilon_h \lim_{K \to \infty} \sum_{k=K_h}^{K} \gamma_k \alpha_k < -\infty, \tag{35}$$

since $\sum_k \alpha_k \gamma_k$ diverges by Assumption 2.1. However, this implies that $z_K < -\infty$ and as a consequence, $d_{K+1} \leq D' + z_K < -\infty$ which is a contradiction as $d_{K+1} \geq 0$. We conclude that $\lim_{k \to \infty} d_k = 0$ and $\lim_{k \to \infty} \bar{x}_k = x^*$, almost surely.

### D.2 Proof of Lemma D.1

The goal is to bound $\|\mathbf{x}_{k+1} - \mathbf{1}\bar{x}_{k+1}\|^2$ by $\|\mathbf{x}_k - \mathbf{1}\bar{x}_k\|^2$ and other vanishing terms.

$$
\begin{aligned}
\|\mathbf{x}_{k+1} - \mathbf{1}\bar{x}_{k+1}\|^2 &= \|\mathbf{x}_{k+1} - \mathbf{1}\bar{x}_k + \mathbf{1}\bar{x}_k - \mathbf{1}\bar{x}_{k+1}\|^2 \\
&= \|\mathbf{x}_{k+1} - \mathbf{1}\bar{x}_k\|^2 + 2\langle \mathbf{x}_{k+1} - \mathbf{1}\bar{x}_k, \mathbf{1}\bar{x}_k - \mathbf{1}\bar{x}_{k+1}\rangle + \|\mathbf{1}\bar{x}_k - \mathbf{1}\bar{x}_{k+1}\|^2 \\
&\stackrel{(a)}{=} \|\mathbf{x}_{k+1} - \mathbf{1}\bar{x}_k\|^2 - \|\mathbf{1}\bar{x}_k - \mathbf{1}\bar{x}_{k+1}\|^2 \\
&\leq \|\mathbf{x}_{k+1} - \mathbf{1}\bar{x}_k\|^2 \\
&= \sum_{i=1}^{n} \|x_{i,k+1} - \bar{x}_k\|^2 \\
&\stackrel{(b)}{\leq} \sum_{i=1}^{n} \|z_{i,k+1} - \bar{x}_k\|^2 \\
&= \|\mathbf{z}_{k+1} - \mathbf{1}\bar{x}_k\|^2 \\
&= \|W\mathbf{x}_k - \alpha_k W\mathbf{g}_k - \mathbf{1}\bar{x}_k\|^2 \\
&= \|W\mathbf{x}_k - \mathbf{1}\bar{x}_k\|^2 - 2\alpha_k \langle W\mathbf{x}_k - \mathbf{1}\bar{x}_k, W\mathbf{g}_k\rangle + \alpha_k^2 \|W\mathbf{g}_k\|^2 \\
&\stackrel{(c)}{\leq} \|W\mathbf{x}_k - \mathbf{1}\bar{x}_k\|^2 + \alpha_k [\frac{1 - \rho_w^2}{2\rho_w^2 \alpha_k} \|W\mathbf{x}_k - \mathbf{1}\bar{x}_k\|^2 + \frac{2\rho_w^2 \alpha_k}{1 - \rho_w^2} \|W\mathbf{g}_k\|^2] + \alpha_k^2 \|W\mathbf{g}_k\|^2 \\
&\stackrel{(d)}{\leq} \rho_w^2 \|\mathbf{x}_k - \mathbf{1}\bar{x}_k\|^2 + \alpha_k [\frac{1 - \rho_w^2}{2\alpha_k} \|\mathbf{x}_k - \mathbf{1}\bar{x}_k\|^2 + \frac{2\rho_w^2 \alpha_k}{1 - \rho_w^2} \|W\mathbf{g}_k\|^2] + \alpha_k^2 \|W\mathbf{g}_k\|^2 \\
&= \frac{1 + \rho_w^2}{2} \|\mathbf{x}_k - \mathbf{1}\bar{x}_k\|^2 + \alpha_k^2 \frac{1 + \rho_w^2}{1 - \rho_w^2} \|W\mathbf{g}_k\|^2 \\
&= \frac{1 + \rho_w^2}{2} \|\mathbf{x}_k - \mathbf{1}\bar{x}_k\|^2 + \alpha_k^2 \frac{1 + \rho_w^2}{1 - \rho_w^2} \|W\mathbf{g}_k - \mathbf{1}\bar{g}_k + \mathbf{1}\bar{g}_k\|^2 \\
&\stackrel{(e)}{=} \frac{1 + \rho_w^2}{2} \|\mathbf{x}_k - \mathbf{1}\bar{x}_k\|^2 + \alpha_k^2 \frac{1 + \rho_w^2}{1 - \rho_w^2} \|W\mathbf{g}_k - \mathbf{1}\bar{g}_k\|^2 + \alpha_k^2 \frac{n(1 + \rho_w^2)}{1 - \rho_w^2} \|\bar{g}_k\|^2
\end{aligned}
$$

$$\leq \frac{1+\rho_w^2}{2}\|\mathbf{x}_k - \mathbf{1}\bar{x}_k\|^2 + \alpha_k^2 \frac{\rho_w^2(1+\rho_w^2)}{1-\rho_w^2}\|\mathbf{g}_k - \mathbf{1}\bar{g}_k\|^2 + \alpha_k^2 \frac{n(1+\rho_w^2)}{1-\rho_w^2}\|\bar{g}_k\|^2$$

$$\overset{(f)}{\leq} \frac{1+\rho_w^2}{2}\|\mathbf{x}_k - \mathbf{1}\bar{x}_k\|^2 + \alpha_k^2 \frac{n(1+\rho_w^2)}{1-\rho_w^2}M. \tag{36}$$

where $(a)$ is by (37), $(b)$ is the projection inequality (5) noting that $\bar{x}_k \in \mathcal{K}$ since $\mathcal{K}$ is a convex set (so projecting it onto $\mathcal{K}$ gives us the same point), $(c)$ is by $-2\epsilon \times \frac{1}{\epsilon}\langle a, b\rangle = -2\langle \epsilon a, \frac{1}{\epsilon}b\rangle \leq \epsilon^2\|a\|^2 + \frac{1}{\epsilon^2}\|b\|^2$ $(d)$ is by Lemma 1.4, $(e)$ is by (38), and $(f)$ is by (39) and (40).

$$\begin{aligned}
2\langle \mathbf{x}_{k+1} - \mathbf{1}\bar{x}_k, \mathbf{1}\bar{x}_k - \mathbf{1}\bar{x}_{k+1}\rangle &= 2\sum_{i=1}^{n}\langle x_{i,k+1} - \bar{x}_k, \bar{x}_k - \bar{x}_{k+1}\rangle \\
&= 2\langle \sum_{i=1}^{n}(x_{i,k+1} - \bar{x}_k), \bar{x}_k - \bar{x}_{k+1}\rangle \\
&= 2\langle n(\bar{x}_{k+1} - \bar{x}_k), \bar{x}_k - \bar{x}_{k+1}\rangle \\
&= -2n\langle \bar{x}_k - \bar{x}_{k+1}, \bar{x}_k - \bar{x}_{k+1}\rangle \\
&= -2n\|\bar{x}_k - \bar{x}_{k+1}\|^2 \\
&= -2\|\mathbf{1}\bar{x}_k - \mathbf{1}\bar{x}_{k+1}\|^2.
\end{aligned} \tag{37}$$

$$\begin{aligned}
\langle W\mathbf{g}_k - \mathbf{1}\bar{g}_k, \mathbf{1}\bar{g}_k\rangle &= \sum_{i=1}^{n}\langle \sum_{j=1}^{n}w_{ij}g_{j,k} - \bar{g}_k, \bar{g}_k\rangle \\
&= \langle \sum_{i=1}^{n}\sum_{j=1}^{n}w_{ij}g_{j,k} - n\bar{g}_k, \bar{g}_k\rangle \\
&= \langle \sum_{j=1}^{n}(\sum_{i=1}^{n}w_{ij})g_{j,k} - n\bar{g}_k, \bar{g}_k\rangle \\
&= \langle \sum_{j=1}^{n}g_{j,k} - n\bar{g}_k, \bar{g}_k\rangle \\
&= 0.
\end{aligned} \tag{38}$$

From Lemma B.1, we know that $\|\bar{g}_k\|^2 \leq M < \infty$ almost surely,

$$\begin{aligned}
\|\mathbf{g}_k - \mathbf{1}\bar{g}_k\|^2 &= \sum_{i=1}^{n}\|g_{i,k} - \frac{1}{n}\sum_{j=1}^{n}g_{j,k}\|^2 \\
&= \sum_{i=1}^{n}\left(\|g_{i,k}\|^2 - 2\langle g_{i,k}, \frac{1}{n}\sum_{j=1}^{n}g_{j,k}\rangle + \|\bar{g}_k\|^2\right) \\
&= \|\mathbf{g}_k\|^2 - 2n\|\bar{g}_k\|^2 + n\|\bar{g}_k\|^2 \\
&= \|\mathbf{g}_k\|^2 - n\|\bar{g}_k\|^2
\end{aligned} \tag{39}$$

Then,

$$\begin{aligned}
\rho_w^2\|\mathbf{g}_k - \mathbf{1}\bar{g}_k\|^2 + n\|\bar{g}_k\|^2 &= \rho_w^2\|\mathbf{g}_k\|^2 + n(1-\rho_w^2)\|\bar{g}_k\|^2 \\
&\leq \rho_w^2 nM + n(1-\rho_w^2)M \\
&= nM.
\end{aligned} \tag{40}$$

1. **Proving** $\lim_{K \to \infty} \sum_{k=0}^{K} \|\mathbf{x}_k - \mathbf{1}\bar{x}_k\|^2 < \infty$, $\lim_{K \to \infty} \sum_{k=0}^{K} \|\mathbf{z}_{k+1} - \mathbf{1}\bar{x}_k\|^2 < \infty$, **and** $\lim_{k \to \infty} \|\mathbf{x}_k - \mathbf{1}\bar{x}_k\|^2 = 0$

   Reconsider (36),

$$
\|\mathbf{x}_{k+1} - \mathbf{1}\bar{x}_{k+1}\|^2 \leq \frac{1 + \rho_w^2}{2}\|\mathbf{x}_k - \mathbf{1}\bar{x}_k\|^2 + \alpha_k^2 \frac{n(1 + \rho_w^2)}{1 - \rho_w^2}M
$$

$$
\|\mathbf{x}_k - \mathbf{1}\bar{x}_k\|^2 \leq \frac{1 + \rho_w^2}{2}\|\mathbf{x}_{k-1} - \mathbf{1}\bar{x}_{k-1}\|^2 + \alpha_{k-1}^2 \frac{n(1 + \rho_w^2)}{1 - \rho_w^2}M \qquad (41)
$$

$$
\cdots
$$

$$
\|\mathbf{x}_1 - \mathbf{1}\bar{x}_1\|^2 \leq \frac{1 + \rho_w^2}{2}\|\mathbf{x}_0 - \mathbf{1}\bar{x}_0\|^2 + \alpha_0^2 \frac{n(1 + \rho_w^2)}{1 - \rho_w^2}M.
$$

   Adding all inequalities in (41), we obtain

$$
\|\mathbf{x}_{k+1} - \mathbf{1}\bar{x}_{k+1}\|^2 \leq -\frac{1 - \rho_w^2}{2}\sum_{l=1}^{k} \|\mathbf{x}_l - \mathbf{1}\bar{x}_l\|^2 + \frac{1 + \rho_w^2}{2}\|\mathbf{x}_0 - \mathbf{1}\bar{x}_0\|^2 + \frac{n(1 + \rho_w^2)}{1 - \rho_w^2}M\sum_{l=0}^{k}\alpha_l^2
$$

   Let $k \to \infty$, then the second and third terms are bounded due to Assumption 2.1. There are then 2 cases: $\sum_l \|\mathbf{x}_l - \mathbf{1}\bar{x}_l\|^2$ either diverges or converges. Assume the validity of the hypothesis *H2*) $\sum_l \|\mathbf{x}_l - \mathbf{1}\bar{x}_l\|^2$ diverges, i.e., $\sum_{l=1}^{\infty} \|\mathbf{x}_l - \mathbf{1}\bar{x}_l\|^2 \to \infty$. This leads to

$$
\|\mathbf{x}_{k+1} - \mathbf{1}\bar{x}_{k+1}\|^2 < -\infty,
$$

   as $-\frac{1 - \rho_w^2}{2} < 0$. However, $\|\mathbf{x}_{k+1} - \mathbf{1}\bar{x}_{k+1}\|^2$ should be positive. Thus, hypothesis *H2* cannot be true and $\sum_l \|\mathbf{x}_l - \mathbf{1}\bar{x}_l\|^2$ converges. Hence, $\lim_{k \to \infty} \|\mathbf{x}_k - \mathbf{1}\bar{x}_k\|^2 = 0$ almost surely.

   Thus, reconsider (36),

$$
\|\mathbf{z}_{k+1} - \mathbf{1}\bar{x}_k\|^2 \leq \frac{1 + \rho_w^2}{2}\|\mathbf{x}_k - \mathbf{1}\bar{x}_k\|^2 + \alpha_k^2 \frac{n(1 + \rho_w^2)}{1 - \rho_w^2}M
$$

$$
\sum_{k=0}^{K}\|\mathbf{z}_{k+1} - \mathbf{1}\bar{x}_k\|^2 \leq \frac{1 + \rho_w^2}{2}\sum_{k=0}^{K}\|\mathbf{x}_k - \mathbf{1}\bar{x}_k\|^2 + \frac{n(1 + \rho_w^2)}{1 - \rho_w^2}M\sum_{k=0}^{K}\alpha_k^2 \qquad (42)
$$

$$
< \infty.
$$

2. **Proving** $\sum_{k=0}^{\infty} \gamma_k \alpha_k \|\mathbf{x}_k - \mathbf{1}\bar{x}_k\| < \infty$

   By induction from (36), we have

$$
\|\mathbf{x}_{k+1} - \mathbf{1}\bar{x}_{k+1}\|^2 \leq \left(\frac{1 + \rho_w^2}{2}\right)^{k+1}\|\mathbf{x}_0 - \mathbf{1}\bar{x}_0\|^2 + \frac{2nM}{1 - \rho_w^2}\sum_{j=0}^{k}\left(\frac{1 + \rho_w^2}{2}\right)^{j+1}\alpha_{k-j}^2. \qquad (43)
$$

   Since $\sqrt{a + b} < \sqrt{a} + \sqrt{b}$,

$$
\|\mathbf{x}_{k+1} - \mathbf{1}\bar{x}_{k+1}\| \leq \left(\frac{1 + \rho_w^2}{2}\right)^{\frac{k+1}{2}}\|\mathbf{x}_0 - \mathbf{1}\bar{x}_0\| + \sqrt{\frac{2nM}{1 - \rho_w^2}}\sum_{j=0}^{k}\left(\frac{1 + \rho_w^2}{2}\right)^{\frac{j+1}{2}}\alpha_{k-j}. \qquad (44)
$$

   Then, substituting into the sum $\sum_{k=0}^{\infty} \gamma_k \alpha_k \|\mathbf{x}_k - \mathbf{1}\bar{x}_k\|$,

$$
\sum_{k=1}^{\infty}\gamma_k\alpha_k\left(\left(\frac{1 + \rho_w^2}{2}\right)^{\frac{k}{2}}\|\mathbf{x}_0 - \mathbf{1}\bar{x}_0\| + \sqrt{\frac{2nM}{1 - \rho_w^2}}\sum_{j=0}^{k-1}\left(\frac{1 + \rho_w^2}{2}\right)^{\frac{j+1}{2}}\alpha_{k-1-j}\right)
$$

$$
\leq \gamma_0\alpha_0\|\mathbf{x}_0 - \mathbf{1}\bar{x}_0\|\frac{\sqrt{1 + \rho_w^2}}{\sqrt{2} - \sqrt{1 + \rho_w^2}} + \sqrt{\frac{2nM}{1 - \rho_w^2}}\sum_{k=1}^{\infty}\gamma_k\alpha_k\sum_{j=0}^{k-1}\left(\frac{1 + \rho_w^2}{2}\right)^{\frac{j+1}{2}}\alpha_{k-1-j},
$$

where the inequality is due to the fact that $\gamma_k$ and $\alpha_k$ are both decreasing step-sizes and we have a geometric sum of ratio $\sqrt{\frac{1+\rho^2}{2}} < 1$. We then study the sums in the second term,

$$
\begin{aligned}
\sum_{k=1}^{\infty} \gamma_k \alpha_k \sum_{j=0}^{k-1} \Big(\frac{1+\rho_w^2}{2}\Big)^{\frac{j+1}{2}} \alpha_{k-1-j} &\leq \sum_{k=1}^{\infty} \gamma_k \sum_{j=0}^{k-1} \Big(\frac{1+\rho_w^2}{2}\Big)^{\frac{j+1}{2}} \alpha_{k-1-j}^2 \\
&= \sum_{k=1}^{\infty} \gamma_k \sum_{j=1}^{k} \Big(\frac{1+\rho_w^2}{2}\Big)^{\frac{k-j+1}{2}} \alpha_{j-1}^2 \\
&= \sum_{j=1}^{\infty} \alpha_{j-1}^2 \sum_{k=j}^{\infty} \gamma_k \Big(\frac{1+\rho_w^2}{2}\Big)^{\frac{k-j+1}{2}} \\
&\leq \gamma_0 \sum_{j=1}^{\infty} \alpha_{j-1}^2 \sum_{k=j}^{\infty} \Big(\frac{1+\rho_w^2}{2}\Big)^{\frac{k-j+1}{2}} \\
&= \gamma_0 \frac{\sqrt{1+\rho_w^2}}{\sqrt{2} - \sqrt{1+\rho_w^2}} \sum_{j=1}^{\infty} \alpha_{j-1}^2 \\
&< \infty,
\end{aligned}
$$

as $\sum \alpha_k^2$ converges by Assumption 2.1.

Finally, $\sum_{k=0}^{\infty} \gamma_k \alpha_k \|\mathbf{x}_k - \mathbf{1}\bar{x}_k\| < \infty$.

## D.3 Convergence Rate of the Consensus Error $\|\mathbf{x}_k - \mathbf{1}\bar{x}_k\|^2$ and of $\|\mathbf{z}_{k+1} - \mathbf{1}\bar{x}_k\|^2$

As $\sum_k \|\mathbf{x}_k - \mathbf{1}\bar{x}_k\|^2 < \infty$, let us assume that $\|\mathbf{x}_k - \mathbf{1}\bar{x}_k\|^2$ vanishes with the same rate as $\alpha_k^2$. Then, there must be a scalar $\vartheta_1 > 0$ such that $\|\mathbf{x}_k - \mathbf{1}\bar{x}_k\|^2 < \vartheta_1^2 \alpha_k^2$. To test if such $\vartheta_1$ exists, we employ (36) to check whether $\|\mathbf{x}_{k+1} - \mathbf{1}\bar{x}_{k+1}\|^2 < \vartheta_1^2 \alpha_{k+1}^2$ holds,

$$
\begin{aligned}
\|\mathbf{x}_{k+1} - \mathbf{1}\bar{x}_{k+1}\|^2 &\leq \frac{1+\rho_w^2}{2} \|\mathbf{x}_k - \mathbf{1}\bar{x}_k\|^2 + \frac{n(1+\rho_w^2)M}{1-\rho_w^2} \alpha_k^2 \\
&\leq \frac{1+\rho_w^2}{2} \vartheta_1^2 \alpha_k^2 + \frac{n(1+\rho_w^2)M}{1-\rho_w^2} \alpha_k^2 \\
&= \Big(\frac{1+\rho_w^2}{2} \vartheta_1^2 + \frac{n(1+\rho_w^2)M}{1-\rho_w^2}\Big) \alpha_k^2.
\end{aligned}
\tag{45}
$$

Then, testing

$$
\begin{aligned}
\Big(\frac{1+\rho_w^2}{2} \vartheta_1^2 + \frac{n(1+\rho_w^2)M}{1-\rho_w^2}\Big) \alpha_k^2 &\leq \vartheta_1^2 \alpha_{k+1}^2 \\
\frac{n(1+\rho_w^2)M}{1-\rho_w^2} &\leq \vartheta_1^2 \Big(\frac{\alpha_{k+1}^2}{\alpha_k^2} - \frac{1+\rho_w^2}{2}\Big) \\
\frac{\frac{n(1+\rho_w^2)M}{1-\rho_w^2}}{\frac{\alpha_{k+1}^2}{\alpha_k^2} - \frac{1+\rho_w^2}{2}} &\leq \vartheta_1^2.
\end{aligned}
\tag{46}
$$

Thus, $0 < \varrho^2 < \infty$ whenever $\frac{\alpha_{k+1}^2}{\alpha_k^2} - \frac{1+\rho_w^2}{2} > 0$.

Let us consider $\alpha_k$ having the form in Example 2.3, then $\frac{\alpha_{k+1}^2}{\alpha_k^2} = \Big(\frac{k+1}{k+2}\Big)^{2\upsilon_1}$ is an increasing function of $k$ taking values between 0 and 1, and define

$$
K_1 = \underset{\frac{\alpha_{k+1}^2}{\alpha_k^2} > \frac{1+\rho_w^2}{2}}{\arg\min} \; k.
$$

To test whether $K_1$ grows very large, we find the intersection $\frac{\alpha_{k+1}^2}{\alpha_k^2} = \frac{1+\rho_w^2}{2}$,

$$
\begin{aligned}
\left(\frac{k+1}{k+2}\right)^{2v_1} &= \frac{1+\rho_w^2}{2} \\
\frac{k+1}{k+2} &= \left(\frac{1+\rho_w^2}{2}\right)^{\frac{1}{2v_1}} \\
k+1 &= (k+2)\left(\frac{1+\rho_w^2}{2}\right)^{\frac{1}{2v_1}} \\
k &= \frac{2\left(\frac{1+\rho_w^2}{2}\right)^{\frac{1}{2v_1}} - 1}{1 - \left(\frac{1+\rho_w^2}{2}\right)^{\frac{1}{2v_1}}}.
\end{aligned}
\tag{47}
$$

Define the function $h(x, v_1) = \frac{2x^{\frac{1}{2v_1}} - 1}{1 - x^{\frac{1}{2v_1}}}$ for $0 < x < 1$ and $0.5 < v < 1$.

$\frac{\partial h(x,v_1)}{\partial v_1} = -\frac{\exp(\frac{\ln x}{2v_1})\ln x}{2v_1^2(1 - x^{\frac{1}{2v_1}})^2} > 0$ for a fixed $0 < x < 1$.

$\frac{\partial h(x,v_1)}{\partial x} = \frac{x^{\frac{-2v_1+1}{2v_1}}}{2v_1(1 - x^{\frac{1}{2v_1}})^2} > 0$ for a fixed $0.5 < v_1 < 1$.

Taking an extreme case of $x = v_1 = 0.99$, we obtain $h(0.99, 0.99) \approx 196$ iterations. For $x = v_1 = 0.95$, $h(0.95, 0.95) \approx 36$ iterations. It decreases even more drastically for realistic choices of $\rho_w$ and $v_1$. Thus, it is reasonable to study the rate for $k \geq K_1$.

We conclude that for $k \geq K_1$, there exists $0 < \vartheta_1 < \infty$, such that

$$
\|\mathbf{x}_k - \mathbf{1}\bar{x}_k\|^2 < \vartheta_1^2 \alpha_k^2.
\tag{48}
$$

Thus, from (36), for $k \geq K_1$, we also have

$$
\begin{aligned}
\|\mathbf{z}_{k+1} - \mathbf{1}\bar{x}_k\|^2 &\leq \frac{1+\rho_w^2}{2}\|\mathbf{x}_k - \mathbf{1}\bar{x}_k\|^2 + \alpha_k^2 \frac{n(1+\rho_w^2)M}{1-\rho_w^2} \\
&\leq \left(\frac{1+\rho_w^2}{2}\vartheta_1^2 + \frac{n(1+\rho_w^2)M}{1-\rho_w^2}\right)\alpha_k^2 \\
&:= \vartheta_2^2 \alpha_k^2.
\end{aligned}
\tag{49}
$$

# E Convergence Rate

Our primary result, stated in the following Lemma, is based on finding a relation between two successive iterations of the expected divergence.

**Lemma E.1.** *Let* $A = \frac{\lambda c_3}{2}$, $B = \frac{4c_3 L^2 \vartheta_1^2}{\lambda n}$, $C = \frac{c_1^2 c_4^6}{c_3 \lambda}$, *and* $E = \frac{\vartheta_2}{n}$. *Then, for* $k > K_1$,

$$
D_{k+1} \leq (1 - A\alpha_k\gamma_k)D_k + B\alpha_k^3\gamma_k + C\alpha_k\gamma_k^3 + E\alpha_k^2.
\tag{50}
$$

*Proof: See Appendix E.2.*

Next, we let

$$
K_2 = \underset{A\alpha_k\gamma_k < 1}{\arg\min} k
$$

and $K_0 = \max\{K_1, K_2\}$. For the ensuing part, the purpose is to locate a vanishing upper bound of $D_k$, making use of the inequality (50). The idea is to propose a decreasing sequence $U_{k+1} \leq U_k$ and suppose that $D_k \leq U_k$, $\forall k \geq K_0$, and then verify that $D_{k+1} \leq U_{k+1}$ by induction. The choice of $U_k$ is the most difficult component as one has to keep in mind the general forms of $\alpha_k$ and $\gamma_k$ in (50) and what kind of decisions to take regarding these forms. An essential property of $U_k$ is presented in the subsequent lemma.

**Lemma E.2.** *If a decreasing sequence $U_{k+1} \leq U_k$ for $k \geq K_0$ exists such that $D_{k+1} \leq U_{k+1}$ can be deduced from $D_k \leq U_k$ and (50), then*

$$U_k \geq \frac{B}{A}\alpha_k^2 + \frac{C}{A}\gamma_k^2 + \frac{E}{A}\frac{\alpha_k}{\gamma_k}. \tag{51}$$

*Proof: See Appendix E.3.*

An important remark is that the lower bound of $U_k$ in (51) is vanishing as $\alpha_k^2, \gamma_k^2$, and $\frac{\alpha_k}{\gamma_k}$ are all vanishing. This lower bound provides an insight on the convergence rate of $D_k$ as it cannot be better than that of $\alpha_k^2, \gamma_k^2$, or $\frac{\alpha_k}{\gamma_k}$.

The previous Lemma allows us to move forward in confirming the existence of the constants $\varsigma_1$ and $\varsigma_2$ that permit $D_k \leq \varsigma_1 \gamma_k^2$ and $D_k \leq \varsigma_2 \frac{\alpha_k}{\gamma_k}$ in Theorem 3.4, respectively.

### E.1 Proof of Theorem 3.4

1. **Proof of** (12)

   By definition of $\varsigma_1$, $D_{K_0} \leq \varsigma_1 \gamma_{K_0}^2$. The next step is to make sure that $D_{k+1} \leq U_{k+1}$ can be obtained from $D_k \leq U_k$, $\forall k \geq K_0$. Take $U_k = \varsigma_1 \gamma_k^2$, let $D_k \leq U_k$ hold, and substitute in (50),

   $$D_{k+1} \leq (1 - A\alpha_k\gamma_k)\varsigma_1\gamma_k^2 + B\alpha_k^3\gamma_k + C\alpha_k\gamma_k^3 + E\alpha_k^2.$$

   We solve $D_{k+1} \leq U_{k+1}$ for $\varsigma_1 \in \mathbb{R}^+$

   $$(1 - A\alpha_k\gamma_k)\varsigma_1\gamma_k^2 + B\alpha_k^3\gamma_k + C\alpha_k\gamma_k^3 + E\alpha_k^2 \leq U_{k+1} = \varsigma_1\gamma_{k+1}^2.$$

   Then, by considering $\kappa_k = \frac{1-(\frac{\gamma_{k+1}}{\gamma_k})^2}{\alpha_k\gamma_k} > 0$ as given in (11),

   $$B\alpha_k^2\gamma_k^{-2} + E\alpha_k\gamma_k^{-3} + C \leq \varsigma_1(A - \kappa_k),$$

   and assuming $A - \kappa_k > 0$, we find a constant $\bar{\varsigma_1}$ such that

   $$\varsigma_1 \geq \bar{\varsigma_1} = \frac{B\alpha_k^2\gamma_k^{-2} + E\alpha_k\gamma_k^{-3} + C}{A - \kappa_k},$$

   keeping in mind that $B\alpha_k^2\gamma_k^{-2} + E\alpha_k\gamma_k^{-3} + C$ is positive by definition. Examine the parameters $\sigma_1$, $\sigma_2$, and $\sigma_3$ as they are introduced in (11), then

   $$\bar{\varsigma_1} \leq \frac{B\sigma_2 + E\sigma_3 + C}{A - \sigma_1},$$

   We conclude that $D_k \leq \varsigma_1 \gamma_k^2$ where $\varsigma_1$ satisfies the definition (13).

2. **Proof of** (14)

   $D_{K_0} \leq \varsigma_2 \frac{\gamma_{K_0}}{\alpha_{K_0}}$ by definition of $\varsigma_2$. $\forall k \geq K_0$, let $D_k \leq \varsigma_2 \frac{\alpha_k}{\gamma_k}$, then

   $$D_{k+1} \leq (1 - A\alpha_k\gamma_k)\varsigma_2\frac{\alpha_k}{\gamma_k} + B\alpha_k^3\gamma_k + C\alpha_k\gamma_k^3 + E\alpha_k^2.$$

   Solving $D_{k+1} \leq \varsigma_2 \frac{\alpha_{k+1}}{\gamma_{k+1}}$ for $\varsigma_2 \in \mathbb{R}^+$,

   $$(1 - A\alpha_k\gamma_k)\varsigma_2\frac{\alpha_k}{\gamma_k} + B\alpha_k^3\gamma_k + C\alpha_k\gamma_k^3 + E\alpha_k^2 \leq \varsigma_2\frac{\alpha_{k+1}}{\gamma_{k+1}}.$$

   Take $\tau_k = \frac{\frac{\alpha_k}{\gamma_k} - \frac{\alpha_{k+1}}{\gamma_{k+1}}}{\alpha_k^2} > 0$ as given in (11), then

   $$B\alpha_k\gamma_k + C\alpha_k^{-1}\gamma_k^3 + E \leq (A - \tau_k)\varsigma_2.$$

If $\frac{\alpha_k}{\gamma_k} - \frac{\alpha_{k+1}}{\gamma_{k+1}} < A\alpha_k^2$, then $\exists \bar{\varsigma}_2$ such that

$$\varsigma_2 \geq \bar{\varsigma}_2 = \frac{B\alpha_k\gamma_k + C\alpha_k^{-1}\gamma_k^3 + E}{(A - \tau_k)}.$$

Examine $\sigma_4, \sigma_5$, and $\sigma_6$ that are defined in (11), we can say

$$\bar{\varsigma}_2 \leq \frac{B\sigma_5 + C\sigma_6 + E}{(A - \sigma_4)}.$$

We conclude that $D_k \leq \varsigma_2 \frac{\alpha_k}{\gamma_k}$ with $\varsigma_2$ satisfying (15).

### E.2   Proof of Lemma E.1

Starting with the same steps as in (27),

$$
\begin{aligned}
D_{k+1} =& \mathbb{E}[\|\bar{x}_{k+1} - x^*\|^2] \\
\leq& \mathbb{E}[\frac{1}{n}\|\mathbf{z}_{k+1} - \mathbf{1}\bar{x}_k\|^2 + 2\langle -\alpha_k\bar{g}_k, \bar{x}_k - x^*\rangle + d_k] \\
=& D_k + \frac{1}{n}\mathbb{E}[\|\mathbf{z}_{k+1} - \mathbf{1}\bar{x}_k\|^2] - 2\alpha_k\mathbb{E}[\langle\bar{x}_k - x^*, \bar{g}_k\rangle] \\
\stackrel{(a)}{=}& D_k + \frac{1}{n}\mathbb{E}[\|\mathbf{z}_{k+1} - \mathbf{1}\bar{x}_k\|^2] - 2c_3\alpha_k\gamma_k\mathbb{E}[\langle\bar{x}_k - x^*, h(\mathbf{x}_k) + \bar{b}_k\rangle] \\
=& D_k + \frac{1}{n}\mathbb{E}[\|\mathbf{z}_{k+1} - \mathbf{1}\bar{x}_k\|^2] - 2c_3\alpha_k\gamma_k\mathbb{E}[\langle\bar{x}_k - x^*, \nabla\mathcal{F}(\bar{x}_k)\rangle] + 2c_3\alpha_k\gamma_k\mathbb{E}[\langle\bar{x}_k - x^*, \nabla\mathcal{F}(\bar{x}_k) - h(\mathbf{x}_k)\rangle] \\
& - 2c_3\alpha_k\gamma_k\mathbb{E}[\langle\bar{x}_k - x^*, \bar{b}_k\rangle]
\end{aligned}
$$

$$(52)$$

where $(a)$ is due to both $\mathbb{E}[e_k|\mathcal{H}_k] = 0$ and (21):

$$
\begin{aligned}
\mathbb{E}[\langle\bar{x}_k - x^*, \bar{g}_k\rangle] &= \mathbb{E}[\langle\bar{x}_k - x^*, \bar{g}_k - \mathbb{E}[\bar{g}_k|\mathcal{H}_k] + \mathbb{E}[\bar{g}_k|\mathcal{H}_k]\rangle] \\
&= \mathbb{E}[\langle\bar{x}_k - x^*, e_k\rangle] + \mathbb{E}[\langle\bar{x}_k - x^*, \mathbb{E}[\bar{g}_k|\mathcal{H}_k]\rangle] \\
&= \mathbb{E}_{\mathcal{H}_k}[\mathbb{E}[\langle\bar{x}_k - x^*, e_k\rangle|\mathcal{H}_k]] + \mathbb{E}[\langle\bar{x}_k - x^*, \mathbb{E}[\bar{g}_k|\mathcal{H}_k]\rangle] \\
&= 0 + \mathbb{E}[\langle\bar{x}_k - x^*, \mathbb{E}[\bar{g}_k|\mathcal{H}_k]\rangle].
\end{aligned}
$$

From Lemma B.1, we have $\mathbb{E}[\|\bar{g}_k\|^2] < \bar{M}$ with $\bar{M}$ a bounded constant.

By the strong convexity in Assumption 1.2, we have

$$
\begin{aligned}
-2c_3\alpha_k\gamma_k\mathbb{E}[\langle\bar{x}_k - x^*, \nabla\mathcal{F}(\bar{x}_k)\rangle] &\leq 2c_3\alpha_k\gamma_k\mathbb{E}[\mathcal{F}(x^*) - \mathcal{F}(\bar{x}_k)] - \lambda c_3\alpha_k\gamma_k\mathbb{E}[\|\bar{x}_k - x^*\|^2] \\
&\leq -\lambda c_3\alpha_k\gamma_k\mathbb{E}[\|\bar{x}_k - x^*\|^2] \\
&= -\lambda c_3\alpha_k\gamma_k D_k,
\end{aligned}
$$

$$(53)$$

where we used the fact that $\mathcal{F}(x^*) - \mathcal{F}(\bar{x}_k) \leq 0$.

Next, from Lemma 1.5, we have

$$
\begin{aligned}
2c_3\alpha_k\gamma_k\langle\bar{x}_k - x^*, \nabla\mathcal{F}(\bar{x}_k) - h(\mathbf{x}_k)\rangle &\leq 2c_3\alpha_k\gamma_k\frac{L}{\sqrt{n}}\|\bar{x}_k - x^*\|\|\mathbf{x}_k - \mathbf{1}\bar{x}_k\| \\
&\stackrel{(a)}{\leq} \frac{\lambda c_3\alpha_k\gamma_k}{4}\|\bar{x}_k - x^*\|^2 + 4c_3\alpha_k\gamma_k\frac{L^2}{\lambda n}\|\mathbf{x}_k - \mathbf{1}\bar{x}_k\|^2,
\end{aligned}
$$

where $(a)$ is due to $2\sqrt{\epsilon} \times \frac{1}{\sqrt{\epsilon}}\langle a, b\rangle = 2\langle\sqrt{\epsilon}a, \frac{1}{\sqrt{\epsilon}}b\rangle \leq \epsilon\|a\|^2 + \frac{1}{\epsilon}\|b\|^2$. From (48), we have for $k \geq K_1$,

$$\|\mathbf{x}_k - \mathbf{1}\bar{x}_k\|^2 \leq \vartheta_1^2\alpha_k^2.$$

Hence,

$$2c_3\alpha_k\gamma_k\mathbb{E}[\langle \bar{x}_k - x^*, \nabla\mathcal{F}(\bar{x}_k) - h(\mathbf{x}_k)\rangle] \leq \frac{\lambda c_3\alpha_k\gamma_k}{4}D_k + \frac{4c_3L^2\vartheta_1^2}{\lambda n}\alpha_k^3\gamma_k. \tag{54}$$

From (22),

$$
\begin{aligned}
-2c_3\alpha_k\gamma_k\mathbb{E}[\langle \bar{x}_k - x^*, \bar{b}_k\rangle] &\leq \frac{\lambda c_3\alpha_k\gamma_k}{4}D_k + \frac{4c_3\alpha_k\gamma_k}{\lambda}\mathbb{E}[\|\bar{b}_k\|^2] \\
&\leq \frac{\lambda c_3\alpha_k\gamma_k}{4}D_k + \frac{c_1^2c_4^6\alpha_k\gamma_k^3}{c_3\lambda}
\end{aligned}
\tag{55}
$$

From (49), for $k \geq K_1$, we have

$$\frac{1}{n}\mathbb{E}[\|\mathbf{z}_{k+1} - \mathbf{1}\bar{x}_k\|^2] \leq \frac{\vartheta_2}{n}\alpha_k^2. \tag{56}$$

Finally, by combining (52), (53), (54), (55), and (56) we get (50).

### E.3  Proof of Lemma E.2

Since $1 - A\alpha_k\gamma_k > 0$ when $k \geq K_0$, we may substitute $D_k \leq U_k$ in (50),

$$D_{k+1} \leq (1 - A\alpha_k\gamma_k)U_k + B\alpha_k^3\gamma_k + C\alpha_k\gamma_k^3 + E\alpha_k^2.$$

Testing $D_{k+1} \leq U_{k+1}$ in the previous inequality, we get

$$(1 - A\alpha_k\gamma_k)U_k + B\alpha_k^3\gamma_k + C\alpha_k\gamma_k^3 + E\alpha_k^2 \leq U_{k+1} \leq U_k$$

$$\frac{B}{A}\alpha_k^2 + \frac{C}{A}\gamma_k^2 + \frac{E}{A}\frac{\alpha_k}{\gamma_k} \leq U_k. \tag{57}$$

### E.4  Proof of Theorem 3.5

Theorem 3.4 indicates that the convergence rate is a function of $\upsilon_1$ and $\upsilon_2$, as $\gamma_k^2 \propto (k+1)^{-2\upsilon_2}$ and $\frac{\alpha_k}{\gamma_k} \propto (k+1)^{-(\upsilon_1-\upsilon_2)}$. Nonetheless, we must still verify the validity of the assumptions presented in the theorem, meaning:

- Are $\sigma_1 < A$ and $\sigma_4 < A$ fulfilled?

- Are $\varsigma_1$ and $\varsigma_2$ bounded?

We must remark that in what follows, the analysis is done for $k \geq K_0$.

Let $\alpha_k$ and $\gamma_k$ have the forms given in (16).

1. **Verifying that $\sigma_1 < A$ and $\sigma_4 < A$**

    The idea is to find a bound on $\alpha_0$ and $\gamma_0$ to guarantee $\sigma_1 < A$ and $\sigma_4 < A$. We start by bounding $\sigma_1$ and $\sigma_4$ from above, i.e.,

    $$\sigma_1 = \max_{k \geq K_0} \frac{1 - (\frac{\gamma_{k+1}}{\gamma_k})^2}{\alpha_k\gamma_k} = \max_{k \geq K_0} \frac{1 - (1 + \frac{1}{k+1})^{-2\upsilon_2}}{\alpha_0\gamma_0(k+1)^{-\upsilon_1-\upsilon_2}}$$

    and

    $$\sigma_4 = \max_{k \geq K_0} \frac{1 - \frac{\alpha_{k+1}\gamma_{k+1}^{-1}}{\alpha_k\gamma_k^{-1}}}{\alpha_k\gamma_k} = \max_{k \geq K_0} \frac{1 - (1 + \frac{1}{k+1})^{-(\upsilon_1-\upsilon_2)}}{\alpha_0\gamma_0(k+1)^{-\upsilon_1-\upsilon_2}}.$$

To do so, we define a function $q(x) = x^{-a}(1 - (1 + x)^{-b})$ with $a, b, x \in (0, 1]$. Since $x^{-a} \leq x^{-1}$, we have $q(x) \leq x^{-1}(1 - (1 + x)^{-b}) = r(x)$. To further bound $q(x)$, We study the derivative of $r(x)$ as it is simpler to do so,

$$r'(x) = x^{-2}\left(((b+1)x + 1)(1 + x)^{-b-1} - 1\right) = x^{-2}s(x).$$

Hence the sign of $r'(x)$ is that of $s(x)$. We again calculate the derivative of $s(x)$ to find its sign,

$$s'(x) = -b(b+1)x(1+x)^{-b-2} \leq 0$$

since $b > 0$ and $x > 0$. Then, $s(x)$ is a decreasing function of $x$ over $(0, 1]$. We remark that $\lim_{x \to 0} s(x) = 0$, meaning $s(x) < 0$ and $r'(x) < 0$, $\forall x \in (0, 1]$. Finally,

$$r(x) < \lim_{x \to 0} r(x) = \frac{1 - (1 + x)^{-b}}{x} = b,$$

and $q(x) \leq r(x) < b$, noting that $\lim_{x \to 0} q(x) = b$ for $a = 1$. We conclude that $\sigma_1 < \frac{2v_2}{\alpha_0 \gamma_0}$ and $\sigma_4 < \frac{v_1 - v_2}{\alpha_0 \gamma_0}$. For $\sigma_1 < A$ and $\sigma_4 < A$ to be valid, we must have

$$\alpha_0 \gamma_0 \geq \max\{2v_2, v_1 - v_2\}/A. \tag{58}$$

2. **Verifying that $\varsigma_1$ and $\varsigma_2$ are bounded**

   The goal is to verify that the constant term in the convergence rate is bounded. Thus, we must check that the lower bounds given in (13) and (15) are indeed finite. We start by analyzing $\sigma_2$ and $\sigma_5$,

   $$\sigma_2 = \alpha_0^2 \gamma_0^{-2} \max_{k \geq K_0} (1 + k)^{-2(v_1 - v_2)} = \alpha_0^2 \gamma_0^{-2}(1 + K_0)^{-2(v_1 - v_2)}, \quad \text{as } 0 < v_2 \leq v_1,$$

   and

   $$\sigma_5 = \alpha_0 \gamma_0 \max_{k \geq K_0} (1 + k)^{-(v_1 + v_2)} = \alpha_0 \gamma_0 (1 + K_0)^{-(v_1 + v_2)}, \quad \text{as } 0 < v_2 + v_1.$$

   We end with the analysis of $\sigma_3$ and $\sigma_6$, i.e.,

   $$\sigma_3 = \alpha_0 \gamma_0^{-3} \max_{k \geq K_0} (1 + k)^{-(v_1 - 3v_2)} = \begin{cases} \alpha_0 \gamma_0^{-3}(1 + K_0)^{-(v_1 - 3v_2)}, & \text{if } v_1 \geq 3v_2, \\ \infty, & \text{if } v_1 < 3v_2, \end{cases}$$

   and

   $$\sigma_6 = \alpha_0^{-1} \gamma_0^3 \max_{k \geq K_0} (1 + k)^{v_1 - 3v_2} = \begin{cases} \alpha_0^{-1} \gamma_0^3(1 + K_0)^{v_1 - 3v_2}, & \text{if } v_1 \leq 3v_2, \\ \infty, & \text{if } v_1 > 3v_2. \end{cases}$$

   There are clearly 3 cases:

   - $v_1 > 3v_2$
     Thus, $\sigma_3$ is bounded.
     Since $\sigma_2$ and $\varsigma_1$ (by definition) are also bounded provided that $\alpha_0 \gamma_0 \geq \frac{2v_2}{A}$ in (58).
     However, $\varsigma_2 \to \infty$ since $\sigma_6 \to \infty$ resulting in a loose upper bound in (14).
     To that end, we can write $D_k \leq \Upsilon_1(1 + k)^{-2v_2}$ with $\Upsilon_1$ a bounded constant.

   - $v_1 < 3v_2$
     Similarly, $\sigma_6$ is bounded while $\sigma_3 \to \infty$. Then, $\exists \, \Upsilon_2 < \infty$, where $D_k \leq \Upsilon_2(1 + k)^{-(v_1 - v_2)}$ provided that $\alpha_0 \gamma_0 \geq \frac{v_1 - v_2}{A}$.

   - $v_1 = 3v_2$
     Both $\sigma_3$ and $\sigma_6$ are bounded allowing both previous inequalities corresponding to $D_k$ to be valid.

By this analysis, we conclude the proof of Theorem 3.5.

We present Figure 8 for easier reading of the conditions on the step sizes' exponents where we plot $v_2$ vs. $v_1$.

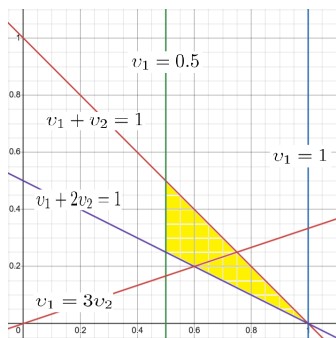

Figure 8: Plot of $v_2$ vs. $v_1$ where the yellow shaded area is the feasibility region determined by Assumption 2.1.

## F  Regret Analysis

Consider the following modified definition of expected divergence that we denote by $D'_k$.

$$D'_k = \mathbb{E}\Big[\frac{1}{n}\sum_{i=1}^{n}\|x_{i,k} - x^*\|^2\Big].$$

We then develop this entity,

$$
\begin{aligned}
D'_{k+1} =& \mathbb{E}\Big[\frac{1}{n}\sum_{i=1}^{n}\|x_{i,k+1} - x^*\|^2\Big] \\
\overset{(a)}{\leq}& \mathbb{E}\Big[\frac{1}{n}\sum_{i=1}^{n}\|z_{i,k+1} - x^*\|^2\Big] \\
=& \mathbb{E}\Big[\frac{1}{n}\sum_{i=1}^{n}\Big\|\sum_{j=1}^{n}w_{ij}(x_{j,k} - \alpha_k g_{j,k}) - x^*\Big\|^2\Big] \\
=& \mathbb{E}\Big[\frac{1}{n}\sum_{i=1}^{n}\Big\|\sum_{j=1}^{n}w_{ij}(x_{j,k} - \alpha_k g_{j,k} - x^*)\Big\|^2\Big] \\
\overset{(b)}{\leq}& \mathbb{E}\Big[\frac{1}{n}\sum_{i=1}^{n}\sum_{j=1}^{n}w_{ij}\|x_{j,k} - \alpha_k g_{j,k} - x^*\|^2\Big] \\
\overset{(c)}{=}& \mathbb{E}\Big[\frac{1}{n}\sum_{j=1}^{n}\|x_{j,k} - \alpha_k g_{j,k} - x^*\|^2\Big] \\
=& D'_k - 2\alpha_k\frac{1}{n}\sum_{j=1}^{n}\mathbb{E}[\langle x_{j,k} - x^*, g_{j,k}\rangle] + \alpha_k^2\frac{1}{n}\sum_{j=1}^{n}\mathbb{E}[\|g_{j,k}\|^2] \\
=& D'_k - 2c_3\alpha_k\gamma_k\frac{1}{n}\sum_{j=1}^{n}\mathbb{E}[\langle x_{j,k} - x^*, \nabla F_j(x_{j,k}) + b_{j,k}\rangle] + \alpha_k^2\frac{1}{n}\sum_{j=1}^{n}\mathbb{E}[\|g_{j,k}\|^2] \\
\overset{(d)}{\leq}& D'_k - 2c_3\alpha_k\gamma_k\frac{1}{n}\sum_{j=1}^{n}\mathbb{E}[F_j(x_{j,k}) - F_j(x^*)] + c_3\alpha_k\gamma_k\frac{1}{n}\sum_{j=1}^{n}\mathbb{E}[\|x_{j,k} - x^*\|^2 + \|b_{j,k}\|^2] + \alpha_k^2 M \\
\overset{(e)}{\leq}& D'_k - 2c_3\alpha_k\gamma_k\frac{1}{n}\sum_{j=1}^{n}\mathbb{E}[F_j(x_{j,k}) - F_j(x^*)] + c_3\alpha_k\gamma_k D'_k + c_3\alpha_k\gamma_k^3\frac{c_4^6 c_1^2}{4c_3^2} + \alpha_k^2 M,
\end{aligned}
$$

where $(a)$ is by applying the projection inequality (5), $(b)$ is by the convexity of the norm square function, $(c)$ is by the doubly stochastic nature of the matrix $W$, $(d)$ is by the convexity of the objective function and Lemma B.1, and $(e)$ is by (20).

Then,

$$\frac{1}{n}\sum_{i=1}^{n}\mathbb{E}[F_i(x_{i,k}) - F_i(x^*)] \leq \frac{D'_k - D'_{k+1}}{c_3\alpha_k\gamma_k} + D'_k + \frac{c_4^6 c_1^2}{4c_3^2}\gamma_k^2 + \frac{\alpha_k}{c_3\gamma_k}M. \tag{59}$$

We know that $D'_k$ can be written as

$$
\begin{aligned}
D'_k &= \mathbb{E}\Big[\frac{1}{n}\sum_{i=1}^{n}\|x_{i,k} - x^*\|^2\Big]\\
&= \mathbb{E}\Big[\frac{1}{n}\sum_{i=1}^{n}\|x_{i,k} - \bar{x}_k + \bar{x}_k - x^*\|^2\Big]\\
&= \mathbb{E}\Big[\frac{1}{n}\sum_{i=1}^{n}\Big(\|x_{i,k} - \bar{x}_k\|^2 + 2\langle x_{i,k} - \bar{x}_k, \bar{x}_k - x^*\rangle + \|\bar{x}_k - x^*\|^2\Big)\Big]\\
&= \mathbb{E}\Big[\frac{1}{n}\|\mathbf{x}_k - \mathbf{1}\bar{x}_k\|^2 + 2\langle \bar{x}_k - \bar{x}_k, \bar{x}_k - x^*\rangle + \|\bar{x}_k - x^*\|^2\Big]\\
&= \mathbb{E}\Big[\frac{1}{n}\|\mathbf{x}_k - \mathbf{1}\bar{x}_k\|^2 + \|\bar{x}_k - x^*\|^2\Big].
\end{aligned}
\tag{60}
$$

Hence, to find the regret bound, we write

$$
\mathbb{E}\Big[\frac{1}{n}\sum_{k=K_0}^{K}\sum_{i=1}^{n}F_i(x_{i,k}) - F_i(x^*)\Big]
$$

$$
\overset{(a)}{\leq} \sum_{k=K_0}^{K}\Big(\frac{D'_k - D'_{k+1}}{c_3\alpha_k\gamma_k} + D'_k + \frac{c_4^6 c_1^2}{4c_3^2}\gamma_k^2 + \frac{\alpha_k}{c_3\gamma_k}M\Big)
$$

$$
= \sum_{k=K_0+1}^{K} D'_k\Big(\frac{1}{c_3\alpha_k\gamma_k} - \frac{1}{c_3\alpha_{k-1}\gamma_{k-1}}\Big) + \frac{D'_{K_0}}{c_3\alpha_{K_0}\gamma_{K_0}} + \frac{D'_{K+1}}{c_3\alpha_{K+1}\gamma_{K+1}} + \sum_{k=K_0}^{K}\Big(D'_k + \frac{c_4^6 c_1^2}{4c_3^2}\gamma_k^2 + \frac{\alpha_k}{c_3\gamma_k}M\Big)
$$

$$
\overset{(b)}{=} \Big(\frac{1}{c_3\alpha_0\gamma_0} + 1\Big)\sum_{k=K_0+1}^{K} D'_k + \Big(\frac{1}{c_3\alpha_{K_0}\gamma_{K_0}} + 1\Big)D'_{K_0} + \frac{D'_{K+1}}{c_3\alpha_{K+1}\gamma_{K+1}} + \sum_{k=K_0}^{K}\Big(\frac{c_4^6 c_1^2\gamma_0^2}{4c_3^2}\frac{1}{\sqrt{k+1}} + \frac{M\alpha_0}{c_3\gamma_0}\frac{1}{\sqrt{k+1}}\Big)
$$

$$
\overset{(c)}{\leq} \Big(\frac{1}{c_3\alpha_0\gamma_0} + 1\Big)\sum_{k=K_0+1}^{K}\Big(\frac{\vartheta_1^2\alpha_0^2}{n}\frac{1}{(k+1)^{\frac{3}{2}}} + \Upsilon\frac{1}{\sqrt{k+1}}\Big) + \frac{(K+2)}{c_3\alpha_0\gamma_0}\Big(\frac{\vartheta_1^2\alpha_0^2}{n}\frac{1}{(K+2)^{\frac{3}{2}}} + \Upsilon\frac{1}{\sqrt{K+2}}\Big)
$$

$$
+ \Big(\frac{1}{c_3\alpha_{K_0}\gamma_{K_0}} + 1\Big)D'_{K_0} + \Big(\frac{c_4^6 c_1^2\gamma_0^2}{4c_3^2} + \frac{M\alpha_0}{c_3\gamma_0}\Big)\sum_{k=K_0}^{K}\frac{1}{\sqrt{k+1}}
$$

where $(a)$ is following up from (59), $(b)$ is by substituting $\alpha_k = \alpha_0(k+1)^{-\frac{3}{4}}$ and $\gamma_k = \gamma_0(k+1)^{-\frac{1}{4}}$, and $(c)$ is by (60), Lemma 3.3, and Theorem 3.5.

To find an upper bound, we interpret the sums over $K_0+1 \leq k \leq K$ as Riemann sums in which the functions $\frac{1}{(u+1)^{\frac{3}{2}}}$ and $\frac{1}{\sqrt{u+1}}$ are evaluated at the right endpoint of the interval $[i-1, i]$ for $i = K_0+1, K_0+2, \ldots, K$. Since the functions $\frac{1}{(u+1)^{\frac{3}{2}}}$ and $\frac{1}{\sqrt{u+1}}$ are monotonically decreasing, the sums are in fact *lower* Riemann sums and therefore bounded from above by the integrals $\int_{K_0}^{K}\frac{1}{(u+1)^{\frac{3}{2}}}du$ and $\int_{K_0}^{K}\frac{1}{\sqrt{u+1}}du$, respectively.

$$\sum_{k=K_0+1}^{K}\frac{1}{(k+1)^{\frac{3}{2}}} \leq \int_{K_0}^{K}\frac{1}{(u+1)^{\frac{3}{2}}}du = 2\Big(\frac{1}{\sqrt{K_0+1}} - \frac{1}{\sqrt{K+1}}\Big)$$

$$\sum_{k=K_0+1}^{K} \frac{1}{\sqrt{k+1}} \leq \int_{K_0}^{K} \frac{1}{\sqrt{u+1}} du = 2(\sqrt{K+1} - \sqrt{K_0+1})$$

Finally,

$$
\begin{aligned}
\mathbb{E}\left[\frac{1}{n}\sum_{k=K_0}^{K}\sum_{i=1}^{n} F_i(x_{i,k}) - F_i(x^*)\right] \leq & 2\left(\frac{1}{c_3\alpha_0\gamma_0}+1\right)\left(\frac{\vartheta_1^2\alpha_0^2}{n}\left(\frac{1}{\sqrt{K_0+1}}-\frac{1}{\sqrt{K+1}}\right)+\Upsilon(\sqrt{K+1}-\sqrt{K_0+1})\right) \\
& + \frac{1}{c_3\alpha_0\gamma_0}\left(\frac{\vartheta_1^2\alpha_0^2}{n}\frac{1}{\sqrt{K+2}}+\Upsilon\sqrt{K+2}\right) \\
& + \left(\frac{1}{c_3\alpha_{K_0}\gamma_{K_0}}+1\right)D'_{K_0} + \left(\frac{c_4^6 c_1^2 \gamma_0^2}{2c_3^2}+\frac{2M\alpha_0}{c_3\gamma_0}\right)(\sqrt{K+1}-\sqrt{K_0})
\end{aligned}
$$

## G Convergence Rate with Constant Step Sizes

We start by going over previous derivations,

$$
\begin{aligned}
\breve{g}_{i,k} &= \mathbb{E}_{S,\Phi,\zeta}[\Phi_{i,k}(f_i(x_{i,k}+\gamma\Phi_{i,k},S_{i,k})+\zeta_{i,k})|\mathcal{H}_k] \\
&= \mathbb{E}_{\Phi}[\Phi_{i,k}F_i(x_{i,k}+\gamma\Phi_{i,k})|\mathcal{H}_k] \\
&= F_i(x_{i,k})\mathbb{E}_{\Phi}[\Phi_{i,k}] + \gamma\mathbb{E}_{\Phi}[\Phi_{i,k}\Phi_{i,k}^T|\mathcal{H}_k]\nabla F_i(x_{i,k}) + \frac{\gamma^2}{2}\mathbb{E}_{\Phi}[\Phi_{i,k}\Phi_{i,k}^T\nabla^2 F_i(\tilde{x}_{i,k})\Phi_{i,k}|\mathcal{H}_k] \\
&= c_3\gamma[\nabla F_i(x_{i,k})+b_{i,k}].
\end{aligned}
$$

Thus, $b_{i,k} = \frac{\gamma}{2c_3}\mathbb{E}_{\Phi}[\Phi_{i,k}\Phi_{i,k}^T\nabla^2 F_i(\tilde{x}_{i,k})\Phi_{i,k}|\mathcal{H}_k]$.

Let Assumptions 1.2 and 2.2 hold. Then, we can bound the bias as

$$
\begin{aligned}
\|b_{i,k}\| &\leq \frac{\gamma}{2c_3}\mathbb{E}_{\Phi}[\|\Phi_{i,k}\|_2\|\Phi_{i,k}^T\|_2\|\nabla^2 F_i(\tilde{x}_{i,k})\|_2\|\Phi_{i,k}\|_2|\mathcal{H}_k] \\
&\leq \gamma\frac{c_4^3 c_1}{2c_3}.
\end{aligned}
$$

We remark that

$$
\begin{aligned}
\tilde{g}_k &= \mathbb{E}[\bar{g}_k|\mathcal{H}_k] \\
&= \frac{1}{n}\sum_{i=1}^{n}\mathbb{E}[g_{i,k}|\mathcal{H}_k] \\
&= \frac{1}{n}\sum_{i=1}^{n} c_3\gamma[\nabla F_i(x_{i,k})+b_{i,k}] \\
&= c_3\gamma[h(\mathbf{x}_k)+\bar{b}_k]
\end{aligned}
\tag{61}
$$

is also a biased estimator of $h(\mathbf{x}_k)$ with

$$
\begin{aligned}
\|\bar{b}_k\| = \|\frac{1}{n}\sum_{i=1}^{n}b_{i,k}\| \\
&\leq \frac{1}{n}\sum_{i=1}^{n}\|b_{i,k}\| \\
&\leq \frac{1}{n}\sum_{i=1}^{n}\gamma\frac{c_4^3 c_1}{2c_3} \\
&= \gamma\frac{c_4^3 c_1}{2c_3}.
\end{aligned}
\tag{62}
$$

**Lemma G.1.** *Let all Assumptions 1.3, 2.2, and 2.4 hold, then there exists a bounded constant $\bar{M} > 0$, such that $E[\|\bar{g}_k\|^2] < \bar{M}$.*

*Proof.* $\forall i \in \mathcal{N}$, we have

$$
\begin{aligned}
\mathbb{E}[\|g_{i,k}\|^2|\mathcal{H}_k] &= \mathbb{E}[\|\Phi_{i,k}(f_i(x_{i,k} + \gamma\Phi_{i,k}, S_{i,k}) + \zeta_{i,k})\|^2|\mathcal{H}_k] \\
&= \mathbb{E}[\|\Phi_{i,k}\|^2\|f_i(x_{i,k} + \gamma\Phi_{i,k}, S_{i,k}) + \zeta_{i,k}\|^2|\mathcal{H}_k] \\
&\stackrel{(a)}{\leq} c_4^2\mathbb{E}[(f_i(x_{i,k} + \gamma\Phi_{i,k}, S_{i,k}) + \zeta_{i,k})^2|\mathcal{H}_k] \\
&\stackrel{(b)}{=} c_4^2\mathbb{E}[f_i^2(x_{i,k} + \gamma\Phi_{i,k}, S_{i,k})|\mathcal{H}_k] + c_4^2 c_2 \\
&\stackrel{(c)}{<} \infty,
\end{aligned}
$$

where $(a)$ is due to Assumption 2.2, $(b)$ Assumption 1.3, and $(c)$ Assumption 2.4. $\qquad\square$

The stochastic noise is still defined as $e_k = \bar{g}_k - \tilde{g}_k$ and retains its property

$$
\mathbb{E}[e_k] = \mathbb{E}[\bar{g}_k - \mathbb{E}[\bar{g}_k|\mathcal{H}_k]] = \mathbb{E}_{\mathcal{H}_k}\left[\mathbb{E}\left[\bar{g}_k - \mathbb{E}[\bar{g}_k|\mathcal{H}_k]\Big|\mathcal{H}_k\right]\right] = 0.
$$

1. **Proving $\|\mathbf{x}_k - \mathbf{1}\bar{x}_k\|^2$ and $\|\mathbf{z}_{k+1} - \mathbf{1}\bar{x}_k\|^2$ converge linearly**

$$
\begin{aligned}
\|\mathbf{x}_{k+1} - \mathbf{1}\bar{x}_{k+1}\|^2 &= \|\mathbf{x}_{k+1} - \mathbf{1}\bar{x}_k + \mathbf{1}\bar{x}_k - \mathbf{1}\bar{x}_{k+1}\|^2 \\
&= \|\mathbf{x}_{k+1} - \mathbf{1}\bar{x}_k\|^2 + 2\langle\mathbf{x}_{k+1} - \mathbf{1}\bar{x}_k, \mathbf{1}\bar{x}_k - \mathbf{1}\bar{x}_{k+1}\rangle + \|\mathbf{1}\bar{x}_k - \mathbf{1}\bar{x}_{k+1}\|^2 \\
&\stackrel{(a)}{=} \|\mathbf{x}_{k+1} - \mathbf{1}\bar{x}_k\|^2 - \|\mathbf{1}\bar{x}_k - \mathbf{1}\bar{x}_{k+1}\|^2 \\
&\leq \|\mathbf{x}_{k+1} - \mathbf{1}\bar{x}_k\|^2 \\
&= \sum_{i=1}^n \|x_{i,k+1} - \bar{x}_k\|^2 \\
&\stackrel{(b)}{\leq} \sum_{i=1}^n \|z_{i,k+1} - \bar{x}_k\|^2 \\
&= \|\mathbf{z}_{k+1} - \mathbf{1}\bar{x}_k\|^2 \\
&= \|W\mathbf{x}_k - \alpha W\mathbf{g}_k - \mathbf{1}\bar{x}_k\|^2 \\
&= \|W\mathbf{x}_k - \mathbf{1}\bar{x}_k\|^2 - 2\alpha\langle W\mathbf{x}_k - \mathbf{1}\bar{x}_k, W\mathbf{g}_k\rangle + \alpha^2\|W\mathbf{g}_k\|^2 \\
&\stackrel{(c)}{\leq} \|W\mathbf{x}_k - \mathbf{1}\bar{x}_k\|^2 + \alpha[\frac{1 - \rho_w^2}{2\rho_w^2\alpha}\|W\mathbf{x}_k - \mathbf{1}\bar{x}_k\|^2 + \frac{2\rho_w^2\alpha}{1 - \rho_w^2}\|W\mathbf{g}_k\|^2] + \alpha^2\|W\mathbf{g}_k\|^2 \\
&\stackrel{(d)}{\leq} \rho_w^2\|\mathbf{x}_k - \mathbf{1}\bar{x}_k\|^2 + \alpha[\frac{1 - \rho_w^2}{2\alpha}\|\mathbf{x}_k - \mathbf{1}\bar{x}_k\|^2 + \frac{2\rho_w^2\alpha}{1 - \rho_w^2}\|W\mathbf{g}_k\|^2] + \alpha^2\|W\mathbf{g}_k\|^2 \\
&= \frac{1 + \rho_w^2}{2}\|\mathbf{x}_k - \mathbf{1}\bar{x}_k\|^2 + \alpha^2\frac{1 + \rho_w^2}{1 - \rho_w^2}\|W\mathbf{g}_k\|^2 \\
&= \frac{1 + \rho_w^2}{2}\|\mathbf{x}_k - \mathbf{1}\bar{x}_k\|^2 + \alpha^2\frac{1 + \rho_w^2}{1 - \rho_w^2}\|W\mathbf{g}_k - \mathbf{1}\bar{g}_k + \mathbf{1}\bar{g}_k\|^2 \\
&\stackrel{(e)}{=} \frac{1 + \rho_w^2}{2}\|\mathbf{x}_k - \mathbf{1}\bar{x}_k\|^2 + \alpha^2\frac{1 + \rho_w^2}{1 - \rho_w^2}\|W\mathbf{g}_k - \mathbf{1}\bar{g}_k\|^2 + \alpha^2\frac{n(1 + \rho_w^2)}{1 - \rho_w^2}\|\bar{g}_k\|^2 \\
&\leq \frac{1 + \rho_w^2}{2}\|\mathbf{x}_k - \mathbf{1}\bar{x}_k\|^2 + \alpha^2\frac{\rho_w^2(1 + \rho_w^2)}{1 - \rho_w^2}\|\mathbf{g}_k - \mathbf{1}\bar{g}_k\|^2 + \alpha^2\frac{n(1 + \rho_w^2)}{1 - \rho_w^2}\|\bar{g}_k\|^2 \\
&\stackrel{(f)}{\leq} \frac{1 + \rho_w^2}{2}\|\mathbf{x}_k - \mathbf{1}\bar{x}_k\|^2 + \alpha^2\frac{n(1 + \rho_w^2)}{1 - \rho_w^2}M.
\end{aligned}
$$

$$\tag{63}$$

where $(a)$ is by (37), $(b)$ is the projection inequality (5) noting that $\bar{x}_k \in \mathcal{K}$ since $\mathcal{K}$ is a convex set (so projecting it onto $\mathcal{K}$ gives us the same point), $(c)$ is by $-2\epsilon \times \frac{1}{\epsilon}\langle a, b \rangle = -2\langle \epsilon a, \frac{1}{\epsilon} b \rangle \leq \epsilon^2 \|a\|^2 + \frac{1}{\epsilon^2}\|b\|^2$ $(d)$ is by Lemma 1.4, $(e)$ is by (38), and $(f)$ is by (39) and (40).

By induction, we have

$$
\begin{aligned}
\|\mathbf{x}_{k+1} - \mathbf{1}\bar{x}_{k+1}\|^2 &\leq \left(\frac{1 + \rho_w^2}{2}\right)^{k+1}\|\mathbf{x}_0 - \mathbf{1}\bar{x}_0\|^2 + \alpha^2 \frac{2nM}{1 - \rho_w^2}\sum_{j=0}^{k}\left(\frac{1 + \rho_w^2}{2}\right)^{j+1} \\
&\leq \left(\frac{1 + \rho_w^2}{2}\right)^{k+1}\|\mathbf{x}_0 - \mathbf{1}\bar{x}_0\|^2 + \alpha^2 \frac{2nM(1 + \rho_w^2)}{(1 - \rho_w^2)^2},
\end{aligned}
\tag{64}
$$

where the last inequality is due to the geometric sum with $\frac{1+\rho_w^2}{2} < 1$

We conclude that $\|\mathbf{x}_k - \mathbf{1}\bar{x}_k\|^2$ converges linearly to an $\alpha^2$ neighborhood of 0, almost surely. Substituting in (63),

$$
\begin{aligned}
\|\mathbf{z}_{k+1} - \mathbf{1}\bar{x}_k\|^2 &\leq \frac{1 + \rho_w^2}{2}\|\mathbf{x}_k - \mathbf{1}\bar{x}_k\|^2 + \alpha^2 \frac{n(1 + \rho_w^2)}{1 - \rho_w^2}M \\
&\leq \frac{1 + \rho_w^2}{2}\left(\left(\frac{1 + \rho_w^2}{2}\right)^{k}\|\mathbf{x}_0 - \mathbf{1}\bar{x}_0\|^2 + \alpha^2 \frac{2nM(1 + \rho_w^2)}{(1 - \rho_w^2)^2}\right) + \alpha^2 \frac{n(1 + \rho_w^2)}{1 - \rho_w^2}M \\
&= \left(\frac{1 + \rho_w^2}{2}\right)^{k+1}\|\mathbf{x}_0 - \mathbf{1}\bar{x}_0\|^2 + \alpha^2 nM\left(\left(\frac{1 + \rho_w^2}{1 - \rho_w^2}\right)^2 + \frac{1 + \rho_w^2}{1 - \rho_w^2}\right)
\end{aligned}
\tag{65}
$$

Finally, $\|\mathbf{z}_{k+1} - \mathbf{1}\bar{x}_k\|^2$ converges linearly to an $\alpha^2$ neighborhood of 0, almost surely, as well.

2. **Proving $D_k = \mathbb{E}[\|\bar{x}_k - x^*\|^2]$ converges linearly**

$$
\begin{aligned}
D_{k+1} &= \mathbb{E}[\|\bar{x}_{k+1} - x^*\|^2] \\
&\leq \mathbb{E}[\frac{1}{n}\|\mathbf{z}_{k+1} - \mathbf{1}\bar{x}_k\|^2 + 2\langle -\alpha\bar{g}_k, \bar{x}_k - x^* \rangle + d_k] \\
&= D_k + \frac{1}{n}\mathbb{E}[\|\mathbf{z}_{k+1} - \mathbf{1}\bar{x}_k\|^2] - 2\alpha\mathbb{E}[\langle \bar{x}_k - x^*, \bar{g}_k \rangle] \\
&\overset{(a)}{=} D_k + \frac{1}{n}\mathbb{E}[\|\mathbf{z}_{k+1} - \mathbf{1}\bar{x}_k\|^2] - 2c_3\alpha\gamma\mathbb{E}[\langle \bar{x}_k - x^*, h(\mathbf{x}_k) + \bar{b}_k \rangle] \\
&= D_k + \frac{1}{n}\mathbb{E}[\|\mathbf{z}_{k+1} - \mathbf{1}\bar{x}_k\|^2] - 2c_3\alpha\gamma\mathbb{E}[\langle \bar{x}_k - x^*, \nabla\mathcal{F}(\bar{x}_k) \rangle] + 2c_3\alpha\gamma\mathbb{E}[\langle \bar{x}_k - x^*, \nabla\mathcal{F}(\bar{x}_k) - h(\mathbf{x}_k) \rangle] \\
&\quad - 2c_3\alpha\gamma\mathbb{E}[\langle \bar{x}_k - x^*, \bar{b}_k \rangle]
\end{aligned}
\tag{66}
$$

where $(a)$ is due to both $\mathbb{E}[e_k|\mathcal{H}_k] = 0$ and (21):

$$
\begin{aligned}
\mathbb{E}[\langle \bar{x}_k - x^*, \bar{g}_k \rangle] &= \mathbb{E}[\langle \bar{x}_k - x^*, \bar{g}_k - \mathbb{E}[\bar{g}_k|\mathcal{H}_k] + \mathbb{E}[\bar{g}_k|\mathcal{H}_k] \rangle] \\
&= \mathbb{E}[\langle \bar{x}_k - x^*, e_k \rangle] + \mathbb{E}[\langle \bar{x}_k - x^*, \mathbb{E}[\bar{g}_k|\mathcal{H}_k] \rangle] \\
&= \mathbb{E}_{\mathcal{H}_k}[\mathbb{E}[\langle \bar{x}_k - x^*, e_k \rangle|\mathcal{H}_k]] + \mathbb{E}[\langle \bar{x}_k - x^*, \mathbb{E}[\bar{g}_k|\mathcal{H}_k] \rangle] \\
&= 0 + \mathbb{E}[\langle \bar{x}_k - x^*, \mathbb{E}[\bar{g}_k|\mathcal{H}_k] \rangle].
\end{aligned}
$$

From Lemma G.1, we have $\mathbb{E}[\|\bar{g}_k\|^2] < \bar{M}$ with $\bar{M}$ a bounded constant.

By the strong convexity in Assumption 1.2, we have

$$
\begin{aligned}
-2c_3\alpha\gamma\mathbb{E}[\langle \bar{x}_k - x^*, \nabla\mathcal{F}(\bar{x}_k) \rangle] &\leq 2c_3\alpha\gamma\mathbb{E}[\mathcal{F}(x^*) - \mathcal{F}(\bar{x}_k)] - \lambda c_3\alpha\gamma\mathbb{E}[\|\bar{x}_k - x^*\|^2] \\
&\leq -\lambda c_3\alpha\gamma\mathbb{E}[\|\bar{x}_k - x^*\|^2] \\
&= -\lambda c_3\alpha\gamma D_k,
\end{aligned}
\tag{67}
$$

where we used the fact that $\mathcal{F}(x^*) - \mathcal{F}(\bar{x}_k) \leq 0$.

Next, from Lemma 1.5, we have

$$2c_3\alpha\gamma\langle \bar{x}_k - x^*, \nabla\mathcal{F}(\bar{x}_k) - h(\mathbf{x}_k)\rangle \leq 2c_3\alpha\gamma\frac{L}{\sqrt{n}}\|\bar{x}_k - x^*\|\|\mathbf{x}_k - \mathbf{1}\bar{x}_k\|$$

$$\overset{(a)}{\leq} \frac{\lambda c_3\alpha\gamma}{4}\|\bar{x}_k - x^*\|^2 + 4c_3\alpha\gamma\frac{L^2}{\lambda n}\|\mathbf{x}_k - \mathbf{1}\bar{x}_k\|^2,$$

where $(a)$ is due to $2\sqrt{\epsilon} \times \frac{1}{\sqrt{\epsilon}}\langle a, b\rangle = 2\langle\sqrt{\epsilon}a, \frac{1}{\sqrt{\epsilon}}b\rangle \leq \epsilon\|a\|^2 + \frac{1}{\epsilon}\|b\|^2$.

In (64) and (65), we let $R = \|\mathbf{x}_0 - \mathbf{1}\bar{x}_0\|^2$, and $G_1 = \frac{2nM(1+\rho_w^2)}{(1-\rho_w^2)^2}$, $G_2 = nM\left(\left(\frac{1+\rho_w^2}{1-\rho_w^2}\right)^2 + \frac{1+\rho_w^2}{1-\rho_w^2}\right)$,

$$\|\mathbf{x}_k - \mathbf{1}\bar{x}_k\|^2 \leq \left(\frac{1+\rho_w^2}{2}\right)^k R + \alpha^2 G_1 \quad \text{and} \quad \|\mathbf{z}_{k+1} - \mathbf{1}\bar{x}_k\|^2 \leq \left(\frac{1+\rho_w^2}{2}\right)^{k+1} R + \alpha^2 G_2. \tag{68}$$

Hence,

$$2c_3\alpha\gamma\mathbb{E}[\langle \bar{x}_k - x^*, \nabla\mathcal{F}(\bar{x}_k) - h(\mathbf{x}_k)\rangle] \leq \frac{\lambda c_3\alpha\gamma}{4}D_k + 4c_3\alpha\gamma\frac{L^2}{\lambda n}\left[\left(\frac{1+\rho_w^2}{2}\right)^k R + \alpha^2 G_1\right]. \tag{69}$$

From (62),

$$-2c_3\alpha\gamma\mathbb{E}[\langle \bar{x}_k - x^*, \bar{b}_k\rangle] \leq \frac{\lambda c_3\alpha\gamma}{4}D_k + \frac{4c_3\alpha\gamma}{\lambda}\mathbb{E}[\|\bar{b}_k\|^2]$$

$$\leq \frac{\lambda c_3\alpha\gamma}{4}D_k + \alpha\gamma^3\frac{c_1^2 c_4^6}{c_3\lambda} \tag{70}$$

Finally, by combining (66), (67), (68), (69), and (70), and setting now $A = \frac{\lambda c_3}{2}$, $B = \frac{4c_3 L^2}{\lambda n}$, and $C = \frac{c_1^2 c_4^6}{c_3\lambda}$, we get

$$D_{k+1} \leq (1 - A\alpha\gamma)D_k + B\alpha\gamma\left[\left(\frac{1+\rho_w^2}{2}\right)^k R + \alpha^2 G_1\right] + \frac{1}{n}\left[\left(\frac{1+\rho_w^2}{2}\right)^{k+1} R + \alpha^2 G_2\right] + C\alpha\gamma^3$$

$$= (1 - A\alpha\gamma)D_k + R\left[B\alpha\gamma + \frac{1}{n}\left(\frac{1+\rho_w^2}{2}\right)\right]\left(\frac{1+\rho_w^2}{2}\right)^k + \alpha^3\gamma B G_1 + \alpha^2\frac{1}{n}G_2 + \alpha\gamma^3 C. \tag{71}$$

Let $\varrho_1 = 1 - A\alpha\gamma$ and $\varrho_2 = \left(\frac{1+\rho_w^2}{2}\right)$. Then, assuming $\alpha\gamma < \frac{1}{A}$ and taking the telescoping sum

$$D_{k+1} \leq \varrho_1^{k+1}D_0 + R\left(B\alpha\gamma + \frac{\varrho_2}{n}\right)\sum_{i=0}^{k}\varrho_1^i\varrho_2^{k-i} + \left(\alpha^3\gamma B G_1 + \alpha^2\frac{1}{n}G_2 + \alpha\gamma^3 C\right)\sum_{i=0}^{k}\varrho_1^i$$

$$= \varrho_1^{k+1}D_0 + R\left(B\alpha\gamma + \frac{\varrho_2}{n}\right)\sum_{i=0}^{k}\varrho_1^i\varrho_2^{k-i} + \left(\alpha^3\gamma B G_1 + \alpha^2\frac{1}{n}G_2 + \alpha\gamma^3 C\right)\left(\frac{1 - \varrho_1^{k+1}}{1 - \varrho_1}\right)$$

$$= \varrho_1^{k+1}D_0 + R\left(B\alpha\gamma + \frac{\varrho_2}{n}\right)\sum_{i=0}^{k}\varrho_1^i\varrho_2^{k-i} + \left(\alpha^2\frac{BG_1}{A} + \frac{\alpha}{\gamma}\frac{G_2}{nA} + \gamma^2\frac{C}{A}\right)(1 - \varrho_1^{k+1})$$

$$\leq \varrho_1^{k+1}D_0 + R\left(B\alpha\gamma + \frac{\varrho_2}{n}\right)\sum_{i=0}^{k}\varrho_1^i\varrho_2^{k-i} + \alpha^2\frac{BG_1}{A} + \frac{\alpha}{\gamma}\frac{G_2}{nA} + \gamma^2\frac{C}{A} \tag{72}$$

where in the last equality, we further imposed the step sizes to satisfy $\alpha < \gamma$.

In what follows, we discuss the summation in the second term of the inequality to avoid setting loose bounds. We know that this summation can be written as follows,

$$\sum_{i=0}^{k}\varrho_1^i\varrho_2^{k-i} = \sum_{i=0}^{k}\varrho_1^{k-i}\varrho_2^i \tag{73}$$

Thus, without imposing further assumptions on the step sizes, we consider the following function the two cases:

- When $\varrho_1 \leq \varrho_2$, we use the left hand side of the previous equality

$$
\begin{aligned}
D_{k+1} &\leq \varrho_1^{k+1} D_0 + R\Big(B\alpha\gamma + \frac{\varrho_2}{n}\Big)\varrho_2^k \sum_{i=0}^{k} \varrho_1^i \varrho_2^{-i} + \alpha^2 \frac{BG_1}{A} + \frac{\alpha}{\gamma}\frac{G_2}{nA} + \gamma^2 \frac{C}{A} \\
&\leq \varrho_1^{k+1} D_0 + R\Big(B\alpha\gamma + \frac{\varrho_2}{n}\Big)\varrho_2^k \frac{1}{1 - \frac{\varrho_1}{\varrho_2}} + \alpha^2 \frac{BG_1}{A} + \frac{\alpha}{\gamma}\frac{G_2}{nA} + \gamma^2 \frac{C}{A} \\
&= \varrho_1^{k+1} D_0 + \varrho_2^{k+1}\frac{2R\Big(B\alpha\gamma + \frac{\varrho_2}{n}\Big)}{2A\alpha\gamma + \rho_w^2 - 1} + \alpha^2 \frac{BG_1}{A} + \frac{\alpha}{\gamma}\frac{G_2}{nA} + \gamma^2 \frac{C}{A}
\end{aligned}
\tag{74}
$$

Then, for arbitrary small step sizes satisfying $\alpha\gamma < \frac{1}{A}$ and $\alpha < \gamma$, $D_k$ converges with the linear rate of $O\big(\varrho_2^k\big)$.

- When $\varrho_1 > \varrho_2$, we use the right hand side

$$
\begin{aligned}
D_{k+1} &\leq \varrho_1^{k+1} D_0 + R\Big(B\alpha\gamma + \frac{\varrho_2}{n}\Big)\varrho_1^k \sum_{i=0}^{k} \varrho^{-i} \varrho_2^i + \alpha^2 \frac{BG_1}{A} + \frac{\alpha}{\gamma}\frac{G_2}{nA} + \gamma^2 \frac{C}{A} \\
&\leq \varrho_1^{k+1} D_0 + R\Big(B\alpha\gamma + \frac{\varrho_2}{n}\Big)\varrho_1^k \frac{1}{1 - \frac{\varrho_2}{\varrho_1}} + \alpha^2 \frac{BG_1}{A} + \frac{\alpha}{\gamma}\frac{G_2}{nA} + \gamma^2 \frac{C}{A} \\
&= \varrho_1^{k+1}\Big(D_0 + \frac{2RB\alpha\gamma + \frac{2R\varrho_2}{n}}{1 - 2A\alpha\gamma - \rho_w^2}\Big) + \alpha^2 \frac{BG_1}{A} + \frac{\alpha}{\gamma}\frac{G_2}{nA} + \gamma^2 \frac{C}{A}
\end{aligned}
\tag{75}
$$

Then, for arbitrary small step sizes satisfying $\alpha\gamma < \frac{1}{A}$ and $\alpha < \gamma$, $D_k$ converges with the linear rate of $O\big(\varrho_1^k\big)$.

# H    Additional Numerical Examples

Figures 9-11 depict the classification of images with the labels 2 and 3 and Figures 12-14 depict those with the labels 3 and 4.

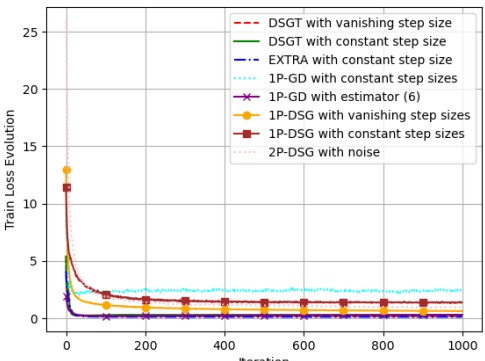

Figure 9: Expected loss function evolution of the proposed algorithm vs. DSGT, EXTRA, and 1P-GD considering vanishing vs. constant step sizes classifying images with labels 2 and 3.

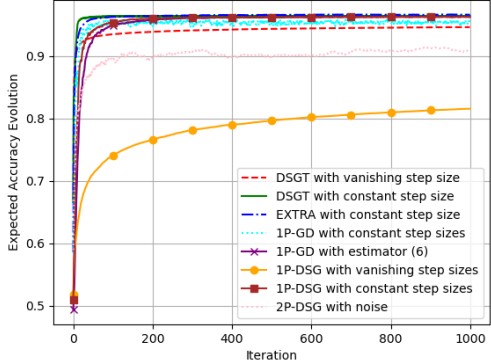

Figure 10: Expected test accuracy evolution of the proposed algorithm vs. DSGT, EXTRA, and 1P-GD considering vanishing vs. constant step sizes classifying images with labels 2 and 3.

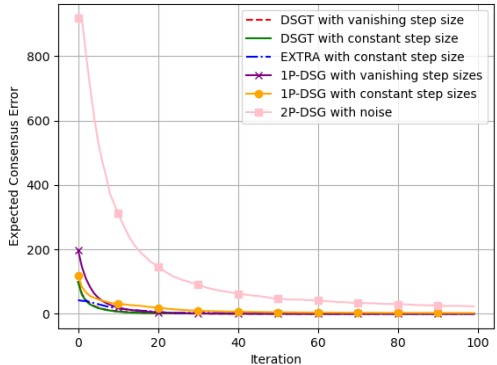

Figure 11: Expected consensus error evolution of the proposed algorithm vs. DSGT and EXTRA considering vanishing vs. constant step sizes classifying images with labels 2 and 3.

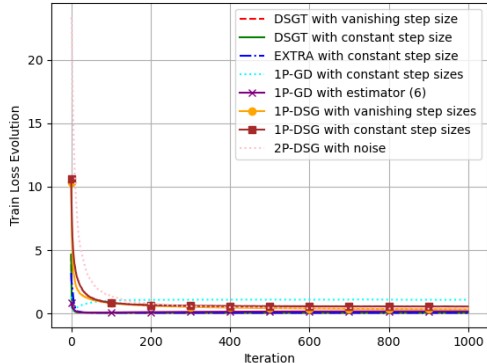

Figure 12: Expected loss function evolution of the proposed algorithm vs. DSGT, EXTRA, and 1P-GD considering vanishing vs. constant step sizes classifying images with labels 3 and 4.

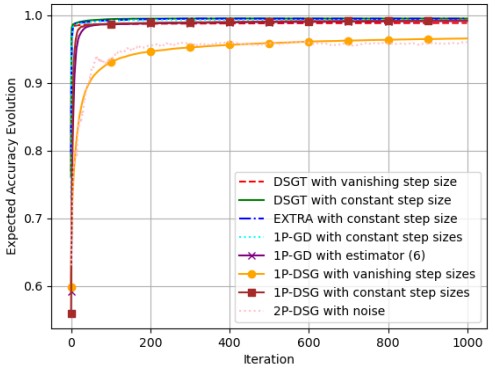

Figure 13: Expected test accuracy evolution of the proposed algorithm vs. DSGT, EXTRA, and 1P-GD considering vanishing vs. constant step sizes classifying images with labels 3 and 4.

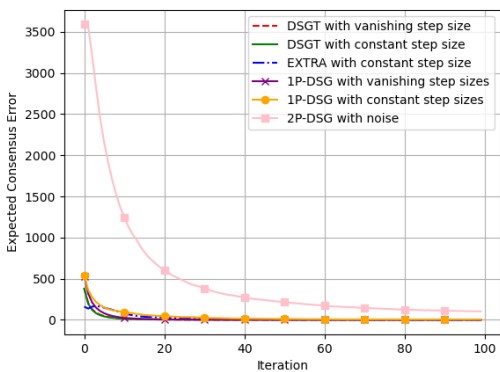

Figure 14: Expected consensus error evolution of the proposed algorithm vs. DSGT and EXTRA considering vanishing vs. constant step sizes classifying images with labels 3 and 4.

