# OpenReview forum: "Zero-Order One-Point Gradient Estimate in Consensus-Based Distributed Stochastic Optimization"
_TMLR — Accepted by TMLR_

### Review · Reviewer_aMVf · 2024-09-24

**Summary Of Contributions:**

The authors consider a distributed stochastic optimization problem under the bandit setting, and propose a method (1P-DSD) with zero-order one-point estimators. They analyze the convergence rate of their method under constant and vanishing step sizes, and provide numerical experiments.

**Audience:**

Yes

**Broader Impact Concerns:**

NULL

**Claims And Evidence:**

Yes

**Requested Changes:**

1) In numerical experiments, it would be better if the authors could compare the proposed method with other zero-order distributed methods such as the 2-point estimate method in Tang et al. (2021)?
2) From Table 1, the Lipschitz condition is required for existing centralized stochastic optimization to achieve the $O(\sqrt{k})$ regret bound with only one-point feedback. By contrast, it seems that this condition is not required by the proposed method. It would be better if the authors could provide more insights on this improvement.

**Strengths And Weaknesses:**

#Strengths
1) The authors have made some efforts to address concerns in the first round review, such as removing the restrictive assumption and replacing the gradient tracking technique.
2) A regret bound of $O(\sqrt{k})$ and a convergence rate of $O(1/\sqrt{k})$ are achieved for smooth and strongly convex functions.
3) A linear rate to $O(\alpha)$-neighborhood of the optimal solution is achieved for the setting with only one-point estimators.

#Weakness
1) The $O(\sqrt{k})$ regret bound and the $O(1/\sqrt{k})$ convergence rate have already been established in the centralized setting. It is not surprising that such results can be extended to the decentralized setting.
2) In the proposed algorithm, i.e., Eq. (7) and Eq. (8), although $ x_{i,k+1}$ is in the feasible set $\mathcal{K}$, the performed point $x_{i,k+1}+\gamma_{k+1}\phi_{i,k+1}$ may not be in the set, which could affect the availability of the one-point feedback.
3) The problem studied in this paper is very close to distributed online bandit optimization (DOBO). Moreover, there do exist some previous studies on DOBO [1, 2] that have established $O(k^{3/4})$ and $O(k^{2/3})$ regret bound for convex and strongly convex functions, respectively. However, the authors did not provide any discussions about these results.
4) In addition, the authors missed an existing study on distributed stochastic optimization with single-point estimation, though it focused on a different type of function.

[1] Projection-free Distributed Online Convex Optimization with $O(\sqrt{T})$ Communication Complexity. In ICML, 2020.
[2] Projection-free Distributed Online Learning with Sublinear Communication Complexity. In JMLR, 2022.
[3] Single Point-Based Distributed Zeroth-Order Optimization with a Non-Convex Stochastic Objective Function. In ICML 2023.

---

> ### Author Response · Authors · 2024-10-17
> **Response (1/2)**
>
> We thank the reviewer for taking the time to carefully read and review our paper.
>
> To answer the weaknesses,
>
> 1-	Transitioning from a centralized to a decentralized setting often results in a slower convergence rate due to factors like noise and variability in local data and consensus rate of the network. Demonstrating a decentralized approach that maintains the same convergence rate as its centralized counterpart is, therefore, a significant result, as it shows that decentralization can be achieved without sacrificing performance.
>
> We emphasize that the primary contribution of this work is demonstrating a linear convergence rate using a single-point gradient estimate in a stochastic, distributed setting—an inherently interesting result. The convergence analysis with vanishing step sizes establishes a foundational framework, which we build upon in the second part to prove a linear convergence rate with constant step sizes.
>
> 2-	In Assumption 2.4, we define the $c_4\gamma_0$-neighborhood of $\mathcal{K}$, $ N_{c_4\gamma_0}(\mathcal{K}) = \\{x\in\mathbb{R}^d | \inf_{a\in\mathcal{K}}\\|x-a\\|<c_4\gamma_0\\}$ and we assume that $|f_i(x, S)| < \infty, \forall x\in N_{c_4\gamma_0}(\mathcal{K}), \forall S, \forall i$.
> As $\\|\Phi_{i,k}\\|\leq c_4$ and $\gamma_k\leq \gamma_0$ for every $k$, this guarantees the availability of the one-point feedback $x_{i,k+1}+\gamma_{k+1}\Phi_{i,k+1}$.
> Another approach would be to project over $(1-c_4\gamma_0)\mathcal{K}$ instead of $\mathcal{K}$ in the algorithm. This would not affect the result.
>
> 3-      We thank you for bringing these references to our attention. We added them to the literature review in the paper. We must note that [1] and [2] cited by the reviewer have a different strategy to deal with the constrained set $\mathcal{K}$. Instead of projection, they solve a linear optimization problem utilizing a surrogate function they propose.
> Thus, their analysis differs from ours in the sense that they have a different structure of the problem, i.e., the evolution of their variables has a different function of the previous iterations compared to ours. Additionally, they have a delayed variable update/communication step, which is interesting for reducing communication costs. A side remark is that, however, during this delay, they aggregate multiple one-point gradient estimates, which can be viewed technically as using a multi-point estimator, but their analysis does not take that into account and deal with it as a one-point estimate instead.
>
> 4- We thank you for bringing it into our attention. We added it to the literature review.
>
> To address the requested changes,
>
> 1-	We thank you for your suggestion. We initially did not add the 2-point method in Tang et al. (2021), as its theoretical guarantees had not been established for strongly convex and smooth functions. However, we are currently running the simulations of that method and we will add them to the final figures.

---

> > ### Author Response · Authors · 2024-10-17
> > **Response (2/2)**
> >
> > 2-	We thank you for this insightful remark. We must point out that on compact sets, continuous functions are bounded. Thus, as generally it is assumed we have continuous gradients (of the objective functions), these gradients are bounded, and this boundedness is equivalent to having Lipschitz continuous objective functions. Hence, in our work, the function is implicitly Lipschitz continuous, but we do not make use of this property. In what follows, we explain the differences between our work and those with one-point feedback in the table with respect to the Lipschitz continuity.
> >
> > Flaxman et al. (2004), develop two bounds for convex functions, one with Lipschitz continuity and one without.  The authors observe that the bound without this property has an "effective Lipschitz constant". They then improve upon this bound with Lipschitz continuity (i.e., the effective constant in the previous bound is substituted by the exact Lipschitz constant) by assuming that the Lipschitz constant is known and that they can substitute its value in the step size. The regret bound thus improves from $k^{\frac56}$ to $k^{\frac34}$.
> > To clarify, the (effective) Lipschitz continuity property is used only to relate the regret established with a projection to $(1-\alpha)\mathcal{K}$ to that within $\mathcal{K}$.
> >
> >
> >  Bach & Perchet (2016), on the other hand, use the Lipschitz continuity to find an upper bound on the variance of the gradient estimate (its expected norm squared). While this property might improve the variance upper bound if the estimate is a noise-free two-point estimate (and thus improves the rate/regret), i.e., making use of the difference of the function values in the estimate, it does not improve the bound for one-point estimates. Using the assumption that the function is bounded on a compact set or that it is Lipschitz continuous for one-point estimates does not change much for the variance and, thus, the regret. The subsequent analysis encompasses the evolution of the expected error and the employment of other tools/properties to improve upon the errors’ upper bound. The Lipschitz continuity is not employed there, as generally, properties like the (strong) convexity and the smoothness (the Lipschitz continuity of the gradient) are more effective in tightening the upper bounds. Our work similarly did not use the Lipschitz continuity as we employed stronger assumptions like strong convexity and smoothness to tighten the bounds.
> >
> >
> > However, with two-point estimates, the situation is different. For example, Agarwal et al. (2010) employ the Lipschitz property multiple times, where it is used to bound the two queried points both in the regret analysis and the variance of the noise-free two-point estimate and it is again used to bound the difference between the loss function and the smoothed version of it (remark, the one-point and two-point gradient estimates are unbiased estimates of a smoothed version of the objective function by Stokes’ theorem). In our work, we use one-point estimates, and we analyze it using Taylor’s expansion and not Stokes’ theorem as is the case in the work of Flaxman et al. (2004) and Agarwal et al. (2010). This explains why the Lipschitz property was not particularly useful in our analysis.

---

### Review · Reviewer_vy8y · 2024-10-01

**Summary Of Contributions:**

This paper addresses a distributed multi-agent stochastic optimization problem. Each agent holds a local objective function that is smooth, strongly convex, and subject to a stochastic process. The goal is for agents to collaboratively optimize the sum of their local functions, despite only having access to noisy function queries at one point at a time, and without access to the gradient. The authors propose a zero-order (ZO) one-point distributed stochastic gradient (1P-DSG) estimator, extending the standard consensus-based distributed stochastic gradient method to a bandit setting where gradients are unavailable. They prove the algorithm converges to the optimum despite the gradient estimate's bias and analyze its convergence rate. With constant step sizes, the method achieves a linear rate, and with vanishing step sizes, it reaches a rate that matches centralized techniques using one-point estimators. A regret bound is also provided.

**Audience:**

Yes

**Broader Impact Concerns:**

None.

**Claims And Evidence:**

Yes

**Requested Changes:**

Major revision:

1. Please highlight the motivations and contributions of the work.

2. Please verify the proposed approach with more practical experiments.

**Strengths And Weaknesses:**

Strengths:

The theoretical analysis of the proposed method is complete.

weaknesses:

1.Please provide some application background of your considered problem settings and the proposed algorithm. The motivations of this work seem to be weak.

2.What is the main contribution of this work. Is it the gradient estimation? If it is, the reviewer suggests the authors highlight the design of the gradient estimation. Please explain the idea, and the sketch the proof involved, and introduce the novelty that is unique in your work. Otherwise, the present paper is just a simple combination of zero-th order optimization and consensus algorithm. The assumptions are strong, and the algorithm framework are commonly seen. The results seem not to be new.

3. The experiments are performed with a toy low-dimensional classification problem. Please compare you algorithm with others in more practical scenarios. In this convergence figures, the linearly-converging algorithms seem to converge at a slower rate than the sublinearly-converging algorithms. This may happen in principle. But meanwhile this implies: the problem is too simple, or the hyperparameters are suboptimal. In either case, it is not good to verify the constant learning rate leads to linear convergence and diminishing learning rate leads to sublinear convergence.

---

> ### Author Response · Authors · 2024-10-17
>
> We thank the reviewer for taking the time to carefully read and review our paper.
>
> To address the weaknesses section:
>
> 1-	The motivation for extending consensus-based distributed optimization to zero-order, one-point estimators is rooted in the growing need for coordination and decision-making among networked agents in scenarios where gradient information is either unavailable or costly to obtain.  This setting has direct applications in fields such as sensor selection for parameter estimation (Liu et al.; 2018), a fundamental problem in smart grids, communication systems, and wireless sensor networks (see references within). The goal is to seek the optimal tradeoff between sensor activations and estimation accuracy. While first-order and second-order algorithms can be used, they involve the calculation of inverse matrices necessary to evaluate the gradient of the cost function, which can be costly and impractical when the observations vary with time. Another example is generating adversarial attacks that cause perturbations to an image that are visually imperceptible to a human but could mislead a classifier [a]. The adversary can be a standard user who does not have access to the inner structure of ML models and can only query the outputs (label or confidence score) for different inputs. This situation occurs when attacking ML cloud services where the model only serves as an API. Due to the black-box property, the optimization of black-box attacks falls into the category of ZO optimization.  In distributed settings, users may collaboratively generate a common perturbation for their own private images without sharing these images. A third example is hyperparameter tuning in distributed settings [b], where the gradient cannot be computed due to the lack of an explicit form of the loss function.
>
> 2-	The main contribution is not the gradient estimation nor the simple combination of ZO optimization with consensus algorithms. The main contribution is the analysis of this setting. This is the first paper to offer almost sure guarantees in this setting and the first paper to prove a linear rate with one-point (or even noisy two-point) estimates is possible. This linear rate has never been proven before nor thought to be attainable due to the information lost with zero-order estimates, especially with one point (query) and two points only. This provides a promising ground for these estimators to be applied to a wide range of problems and invites further enhancement to their properties and applicability.
>
> We must add that we don't use the standard one-point gradient estimate (1) that is analyzed using Stokes' theorem (Flaxman et al.; 2004), where they prove the estimate is an unbiased estimator of a smoothed version of the function. Instead, we analyze our estimate using Taylor’s theorem, establishing a direct relationship between the expectation of the estimate and the exact gradient. In this relationship, we keep $\gamma$ as a factor multiplied by the exact gradient, we do not cancel by $\gamma$ (i.e., we don’t divide by $\gamma$ in the estimate to remove the factor that appears in Taylor’s expansion). At a first glance, multiplying the gradient by a small/vanishing factor may not seem to work, but since this estimate is part of an optimization algorithm that aims to nullify the gradient’s norm (reach a critical point), the problem of the analysis becomes finding a possible value/rate of decrease of the $\gamma$ factor that permits keeping the goal of nullifying the gradient’s norm. This is done in our work using tools from stochastic approximation techniques, but the idea itself is not standard.
>
> 3-	Thank you for your remark. We must point out that the rate of the linearly converging algorithm is not slower as its curvature is more defined than that of the  $O(\\frac{1}{\\sqrt{k}})$ converging algorithm.
> However, the final gap between the ZO methods is due to converging to an $O(\alpha)$-neighborhood of the optimal solution. Linear algorithms generally do not reach the exact optimal solution even with FO information, due to the non-vanishing step sizes. And the gap between our ZO algorithms and their FO counterparts is due to the perturbations and biasedness of the ZO information.
>
> Given the time constraints, we are prioritizing the simulations requested by the reviewers to compare our algorithm with others over adding new application examples.
>
> [a] W. Fang, Z. Yu, Y. Jiang, Y. Shi, C. N. Jones and Y. Zhou, "Communication-Efficient Stochastic Zeroth-Order Optimization for Federated Learning," in IEEE Transactions on Signal Processing, vol. 70, pp. 5058-5073, 2022, doi: 10.1109/TSP.2022.3214122.
>
> [b] Dai, Z., Low, B. K. H., & Jaillet, P. (2020). Federated Bayesian Optimization via Thompson Sampling. In Advances in Neural Information Processing Systems (Vol. 33, pp. 9687–9699). Available from https://proceedings.neurips.cc/paper_files/paper/2020/file/6dfe08eda761bd321f8a9b239f6f4ec3-Paper.pdf

---

> > ### Comment · Reviewer_vy8y · 2024-11-09
> > **accept**
> >
> > Thank you for you response. My concerns have largely been addressed, and I would like to recommmend acceptance for the paper.

---

### Review · Reviewer_WN2N · 2024-10-04

**Summary Of Contributions:**

In this paper, the authors propose a novel zero-order (ZO) one-point gradient estimate for the distributed stochastic gradient (DSG) method, along with the corresponding 1P-DSG algorithm, which provably converges under certain assumptions. The paper provides convergence rates and regret bounds for the 1P-DSG algorithm under both constant and diminishing step-size choices. With the novel one-point ZO gradient estimation, the authors match the convergence guarantees of the centralized counterpart of ZO methods. They validate the applicability of the proposed method through numerical experiments, demonstrating that it can perform comparably to its centralized and first-order (FO) counterparts.

**Audience:**

Yes

**Broader Impact Concerns:**

N/A, mostly theoretical work

**Claims And Evidence:**

Yes

**Requested Changes:**

* Please provide some clarification/justification for the first point and last two points in Weaknesses.

* Please add the error bars and additional baseline mentioned in the second and third points of Weaknesses

* For minor changes needed, refer to Minor Weaknesses.

**Strengths And Weaknesses:**

**Strengths**

* The problem addressed in the paper is clearly defined and strongly motivated.

* The solution method is explained in a straightforward and intuitive manner, making it relatively easy to follow.

* The paper offers convergence guarantees for the one-point ZO DSG method, which aligns with those of the centralized version, presenting a seemingly novel and interesting result.

**Weaknesses**

* In the comparison of SOTA methods, it is shown that for SOTA centralized one-point ZO analysis, $\mathcal{C}_{lip}$ is used. However, in the proposed distributed ZO analysis, this assumption is not needed. It is unclear whether this need for weaker assumptions comes at the cost of a different set of assumptions or if the provided analysis can improve the centralized analysis by removing existing assumptions. It would be insightful to elaborate on this disparity between the assumptions needed for the existing SOTA centralized one-point ZO analysis and those provided in this work for its decentralized counterpart.

* Given that the proposed algorithm involves significant randomness at several levels, it might be useful to add error bars to show the variance of the proposed method compared to that of other methods.

* For completeness, it seems necessary to add a centralized version of 1P-DSG with the proposed ZO gradient estimate as a baseline. This would provide a fairer and more informative comparison to demonstrate the validity of the proposed one-point gradient estimate compared to the performance of the SOTA one-point gradient estimate in (1).

* It is unclear in the main text how the convergence of the proposed method can be ensured without the normalizing constant $1/\gamma$ used in the existing one-point ZO gradient estimate. While the details can be deferred to the proof, it would be insightful to describe the technical innovation, compared to existing analysis, that allows for establishing convergence without $1/\gamma$.

* The authors should justify (e.g. provide citations) the statement "the good balance between exploitation and exploration" of the constant stepsize version of a gradient method (pages 12 (end) and 13 (beginning)).

**Minor Weaknesses**

* In Table 1, line 2, is $\mathcal{C}_{sm}$ a typo?

---

> ### Author Response · Authors · 2024-10-17
> **Response (1/2)**
>
> We thank the reviewer for taking the time to carefully read and review our paper.
>
> To address the weaknesses section,
>
> 1-	We thank you for this remark, as we already have answered Reviewer aMVF, our assumptions already imply Lipschitz continuity.
> On compact sets, continuous functions are bounded. Thus, since we assumed continuous gradients (of the objective functions), these gradients are bounded, and this boundedness is equivalent to having Lipschitz continuous objective functions. Hence, in our work, the function is implicitly Lipschitz continuous, but we do not make use of this property. To explain the usefulness of the Lipschitz property in the centralized literature, as we already mentioned in our answer to Reviewer aMVF,
>
> Flaxman et al. (2004) use the (effective) Lipschitz continuity property only to relate the regret established with a projection to $(1-\alpha)\mathcal{K}$ to that within $\mathcal{K}$. In our case, we project directly to $\mathcal{K}$ since we assume in Assumption 2.4 that the function is bounded in a $c_4 \gamma_0$ neighborhood of $\mathcal{K}$ which allows the queried point in the estimate to stay feasible.
>
>  Bach & Perchet (2016), on the other hand, use the Lipschitz continuity to find an upper bound on the variance of the gradient estimate (its expected norm squared). While this property might improve the variance upper bound if the estimate is a noise-free two-point estimate (and thus improves the rate/regret), i.e., making use of the difference of the function values in the estimate, it does not improve the bound for one-point estimates. Using the assumption that the function is bounded on a compact set or that it is Lipschitz continuous for one-point estimates does not change much for the variance and, thus, the regret. The subsequent analysis encompasses the evolution of the expected error and the employment of other tools/properties to improve upon the errors’ upper bound. The Lipschitz continuity is not employed there, as generally, properties like the (strong) convexity and the smoothness (the Lipschitz continuity of the gradient) are more effective in tightening the upper bounds. Our work similarly did not use the Lipschitz continuity as we employed stronger assumptions like strong convexity and smoothness to tighten the bounds.
>
> However, with two-point estimates, the situation is different. For example, Agarwal et al. (2010) employ the Lipschitz property multiple times, where it is used to bound the two queried points both in the regret analysis and the variance of the noise-free two-point estimate and it is again used to bound the difference between the loss function and the smoothed version of it (remark, the one-point and two-point gradient estimates are unbiased estimates of a smoothed version of the objective function by Stokes’ theorem). In our work, we use one-point estimates, and we analyze it using Taylor’s expansion and not Stokes’ theorem as is the case in the work of Flaxman et al. (2004) and Agarwal et al. (2010). This explains why the Lipschitz property was not particularly useful in our analysis.
>
> 2-	We thank you for this remark. While we are working on adding such error bars in the simulations, we must point out that in our paper, we prove the almost sure convergence. Thus, convergence is not only a question of expectation. The simulation converges almost surely to the optimum in every single instance, not just on average.
>
> 3-	We thank you for this suggestion, we are working on adding a centralized version of 1P-DSG with the same estimator.
>
> 4- In Flaxman et al. (2004), they use Stokes’ theorem to prove that the one-point estimate (1) is an unbiased estimator of a smoothed version of the function. In our work, we analyze our gradient estimate using Taylor’s theorem, highlighting a direct relationship between the expectation of the estimate and the exact gradient. In this relationship, we keep $\gamma$ as a factor multiplied by the exact gradient, we do not cancel by $\gamma$ (i.e., we don’t divide by $\gamma$ in the estimate to remove the factor that appears in Taylor’s expansion). At a first glance, multiplying the gradient by a small/vanishing factor may not seem to work, but since this estimate is part of an optimization algorithm that aims to nullify the gradient’s norm (reach a critical point), the problem of the analysis becomes finding a possible value/rate of decrease of the $\gamma$ factor that permits keeping the goal of nullifying the gradient’s norm. This is in line with the fundamental idea of stochastic approximation methods. The idea is that the information about the exact gradient direction is not lost by not dividing by $\gamma$; it is just we were able to track this information by clearly defining a relationship with it. In other words, employing a scaled version of the exact gradient (adding to the bias, of course) does not cause the gradient method to diverge when this scale (step size) is well chosen.

---

> ### Author Response · Authors · 2024-10-17
> **Response (2/2)**
>
> 5-	Thank you for your remark. Empirically, as evident in our graphs, it is shown that constant step sizes seem to perform better in the accuracy over the test set, i.e., it generalizes better than the version with vanishing step sizes. The statement we made is an intuitive attempt at explaining these results that seemed consistent with FO and ZO information. The idea behind it is that constant step sizes might help increase the exploration in the parameter space of the generalized loss function (not just the function traced by the empirical gradient computed for the training dataset only) whereas the decreasing step sizes slow down over time, reducing the rate at which the model explores new areas in the generalized parameter space and continues to only "exploit" the direction of the gradient computed for the training set.
> However, for more rigor, we remove this statement and provide the following theoretical result from the literature. For example, in [*], it is proven theoretically that the bound on the generalization error for strongly convex objectives is smaller when the step sizes are constant (theorem 3.9) than when they are vanishing (theorem 3.10), wherein the latter, there is an extra element containing the supremum of the function. Naturally, the step sizes seem to play a role in affecting the bound on the evolution of iterates, which in turn affects the uniform stability of the SGD method (stable means that the loss function is not affected much if one datapoint is different) and the generalization error of an SGD-trained model is upper bounded by the uniform stability bound.
>
> 6-	Thank you for pointing this out. Yes, this is a typo.
>
> [*] Hardt, M., Recht, B., & Singer, Y. (2016). Train faster, generalize better: Stability of stochastic gradient descent. arXiv preprint arXiv:1509.01240. Available from https://arxiv.org/abs/1509.01240

---

### Decision · Action_Editor_MV66 · 2024-11-14

**Recommendation:** Accept as is

**Comment:**

The paper is a significant improvement of the previous TMLR submission with the title "Zero-Order One-Point Estimate with Distributed Stochastic Gradient Techniques". The paper addresses all the concerns on the previous submission, such as the removal of restrictive assumptions $\nabla \mathcal{F}(w^*)=0$, the removal of the unrequired gradient track technique and the improvement of the presentation. All the reviewers are satisfied with the current version. The theoretical and empirical analysis is convincing. Therefore, I would like to recommend the acceptance of the paper for publication.

**Audience:**

The proposed method is useful for large-scale machine learning problems where the gradient information is difficult to obtain. The results should be interesting to TMLR audience.

**Claims And Evidence:**

The paper presents a zero-order one-point estimate for distributed stochastic gradient methods with multiple agents in a noisy setting. This leads to a zero-order 1P-DSG algorithm. The paper presents both theoretical analysis and empirical verifications to show the effectiveness of the algorithms. The analysis shows that the proposed method behaves comparably well to its centralized and first-order counterparts.